# Pan-genome and resistome analysis of extended-spectrum ß-lactamase-producing *Escherichia coli*: A multi-setting epidemiological surveillance study from Malaysia

**Jacky Dwiyanto**[1]*, **Jia Wei Hor**[2], **Daniel Reidpath**[3,4], **Tin Tin Su**[4,5], **Shaun Wen Huey Lee**[6], **Qasim Ayub**[1,7,8], **Faizah Binti Mustapha**[9], **Sui Mae Lee**[1], **Su Chern Foo**[1,8], **Chun Wie Chong**[6,10], **Sadequr Rahman**[1,8]*

1 School of Science, Monash University Malaysia, Bandar Sunway, Malaysia, 2 Department of Medicine, University of Malaya, Kuala Lumpur, Malaysia, 3 Health System and Population Studies Division, International Centre for Diarrhoeal Disease Research, Bangladesh, Dhaka, Bangladesh, 4 South East Asia Community Observatory, Segamat, Malaysia, 5 Jeffrey Cheah School of Medicine and Health Sciences, Monash University Malaysia, Bandar Sunway, Malaysia, 6 School of Pharmacy, Monash University Malaysia, Bandar Sunway, Malaysia, 7 Monash University Malaysia Genomics Facility, Bandar Sunway, Malaysia, 8 Tropical Medicine and Biology Multidisciplinary Platform, Monash University Malaysia, Bandar Sunway, Malaysia, 9 Department of Pathology, Hospital Segamat, Segamat, Malaysia, 10 Institute for Research, Development and Innovation, International Medical University, Kuala Lumpur, Malaysia

* jacky.dwiyanto@monash.edu (JD); sadequr.rahman@monash.edu (SR)

**Data Availability Statement:** Raw data have been uploaded to NCBI public repository with the BioProject No. PRJNA752611 (https://www.ncbi.nlm.nih.gov/bioproject/PRJNA752611).

## Abstract

### Objectives

This study profiled the prevalence of extended-spectrum ß-lactamase-producing *Escherichia coli* (ESBL-EC) in the community and compared their resistome and genomic profiles with isolates from clinical patients through whole-genome sequencing.

### Methods

Fecal samples from 233 community dwellers from Segamat, a town in southern Malaysia, were obtained between May through August 2018. Putative ESBL strains were screened and tested using antibiotic susceptibility tests. Additionally, eight clinical ESBL-EC were obtained from a hospital in the same district between June through October 2020. Whole-genome sequencing was then conducted on selected ESBL-EC from both settings (n = 40) for pan-genome comparison, cluster analysis, and resistome profiling.

### Results

A mean ESBL-EC carriage rate of 17.82% (95% CI: 10.48%– 24.11%) was observed in the community and was consistent across demographic factors. Whole-genome sequences of the ESBL-EC (n = 40) enabled the detection of multiple plasmid replicon groups (n = 28), resistance genes (n = 34) and virulence factors (n = 335), with no significant difference in

**Funding:** This study was funded by the 2017 Monash Malaysia Strategic Large Grant Scheme (LG-2017-01-SCI) to LSM and the grant FRGS/1/2019/SKK01/MUSM/01/1 to SR from the Ministry of Higher Education, Malaysia. The funders had no role in study design, data collection and analysis, decision to publish, or preparation of the manuscript.

**Competing interests:** The authors have declared that no competing interests exist.

the number of genes carried between the community and clinical isolates (plasmid replicon groups, p = 0.13; resistance genes, p = 0.47; virulence factors, p = 0.94). Virulence gene marker analysis detected the presence of extraintestinal pathogenic *E. coli* (ExPEC), uropathogenic *E. coli* (UPEC), and enteroaggregative *E. coli* (EAEC) in both the community and clinical isolates. Multiple $bla_{CTX-M}$ variants were observed, dominated by $bla_{CTX-M-27}$ (n = 12), $bla_{CTX-M-65}$ (n = 10), and $bla_{CTX-M-15}$ (n = 9). The clinical and community isolates did not cluster together based on the pan-genome comparison, suggesting isolates from the two settings were clonally unrelated. However, cluster analysis based on carried plasmids, resistance genes and phenotypic susceptibility profiles identified four distinct clusters, with similar patterns between the community and clinical isolates.

## Conclusion

ESBL-EC from the clinical and community settings shared similar resistome profiles, suggesting the frequent exchange of genetic materials through horizontal gene transfer.

## Introduction

The Centers for Disease Control and Prevention (CDC) has classified extended-spectrum ß-lactamase (ESBL) expression in Enterobacteriaceae as a serious threat to public health due to limited therapeutic options and challenges in controlling its transmission [1]. The surveillance of ESBL is complicated by the commensal and hardy nature of Enterobacteriaceae, where ESBL has been reported not only from the clinical setting but also in asymptomatic community dwellers [2–4], wastewater [5, 6], farm animals and pets [7, 8], and even natural environments [9, 10]. Notably, these nonclinical settings often lack regular antibiotic surveillance and monitoring, rendering them reservoirs for ESBL and other antibiotic resistance genes which can potentially supply these resistance determinants to virulent and pathogenic strains.

The successful propagation of ESBL genes has been linked to the hypervirulent strain *Escherichia coli* ST131 [11]. Since its emergence in the late 2000s [12, 13], *E. coli* ST131 gradually became a major strain causing extraintestinal infections worldwide (e.g., the dominance of ST131 among isolates causing bacteremia in Southeast Asia [11]). Its rapid emergence is driven by the successful acquisition of various virulence factors associated with extraintestinal pathogenic *E. coli* (ExPEC), such as the *iutA* aerobactin receptor and *papG* P fimbrial adhesin virulence genes [14]. Its role in disseminating ESBL lies in its frequent carriage of plasmid groups carrying the $bla_{CTX-M}$ gene, which is frequently co-carried with other resistance genes, particularly aminoglycosides [15, 16]. Additionally, ST131 is also frequently associated with fluoroquinolone resistance, either through the carriage of plasmid-mediated quinolone resistance (PMQR) genes such as *qnrS* or quinolone resistance determining region (QRDR) chromosomal mutations, such as *gyrA* and *parC* [14, 17]. Nevertheless, ESBL dissemination can also be carried and disseminated by commensal strains through horizontal gene transfer of plasmids carrying the ESBL gene [18], as observed in community studies (e.g., [2, 7]).

Regardless of their transmission method, multiple studies have reported the direct transmission of ESBL genes from hospitals into the community [19]. Crucially, the intrafamilial transmission of ESBL genes has also been reported [20], suggesting the ease of transmission of extended-spectrum ß-lactamase-producing *Escherichia coli* (ESBL-EC) among individuals living in close proximity. Moreover, the persistence and stable inheritance of plasmids carrying

ESBL genes, even in the absence of antibiotic selection pressure, has led to the widespread prevalence of ESBL genes worldwide [21].

Southeast Asia is a high-risk region for ESBL colonization, with multiple studies reporting ESBL-EC colonization of individuals after visiting the region [22, 23]. Additionally, the Southeast Asian communities have reported some of the highest ESBL colonization rates globally, reaching up to 75.1% [24]. Despite the known endemicity of ESBLs in the Southeast Asian community, there is a lack of comparative genomic analysis of community and clinical isolates, resulting in a gap in our understanding of the relationship between the commensal ESBL-producing isolates in the community and those causing extraintestinal infections in the region. Unveiling such a link is necessary to inform proper surveillance and antibiotic regulation policies to curb the further spread of ESBL in the region.

In Malaysia, ESBL colonization is often reported in the clinical setting [25–28], mainly in studies of a large regional cohort [29–31], with some published articles on farm animals [32], foods [33], and the environment [6, 34]. A comparison between ESBL-producing *Klebsiella pneumoniae* isolated from a swine and a clinical patient was recently reported [35]. However, the epidemiology of the community-acquired ESBL is largely unknown, as there is a lack of community-based carriage studies. We aimed to address this gap by determining the colonization rate of ESBL-EC from community dwellers in Malaysia. Fecal samples obtained from a community cohort in Segamat, Malaysia, were screened for ESBL-EC. Clinical ESBL-EC isolates from the local hospital were also procured. Afterwards, the community and clinical isolates were compared based on their genome and resistomes through whole-genome sequencing.

## Methods

### Community recruitment

The community recruitment protocol for this study has been described before [36]. Briefly, this study involved the community residents of Segamat, a district located in southern Johor state in peninsular Malaysia. From May through August 2018, independent fecal samples were obtained from 233 community dwellers from 110 households. A written consent form was obtained from each participant. Individuals below the age of 18 provided their written consent forms together with their guardians. This study was approved by the Monash University Human Research Ethics Committee (MUHREC) project number 1516, which adheres to the Declaration of Helsinki.

### Isolation of 3GCR-resistant *Escherichia coli* from the community

Within 24 h of expulsion, each fecal sample was diluted 1:10 in buffered peptone water (Oxoid) and spread plated on MacConkey agar (Oxoid) laced with two mg/L cefotaxime (Gold Biotechnology), and then incubated overnight at 37°C. From each sample, one presumptive *E. coli* isolate was randomly picked using Harrison's disk method. The identity was confirmed through their signature metallic green sheen morphology on Eosin Methylene Blue Agar.

### Phenotypic profiling of ESBL-producing *Escherichia coli* from the community

The ESBL phenotypic profile of the isolates was determined with the combination disk test according to the Clinical Laboratory and Standards Institute (CLSI) 2018 guidelines [37]. Briefly, an isolate was regarded as an ESBL producer if the inhibition zone of either cefotaxime (CTX) or ceftazidime (CAZ) (30 μg, Oxoid) combined with clavulanic acid (1 μg/mL) was ≥5

mm compared to CTX or CAZ without clavulanic acid. The phenotypic profiles of *Klebsiella pneumoniae* ATCC700603 and *E. coli* ATCC25922 were used as the positive and negative control, respectively. Afterwards, the antibiotic susceptibility profiles were determined using disk diffusion test against aminoglycoside (amikacin, AK30), ß-lactam-inhibitors combination (ampicillin-sulbactam, SAM20; piperacillin-tazobactam, TZP110), carbapenem (imipenem, IMP10), 1st and 4th-generation cephalosporin (cefazolin, KZ30; cefepime, FEP30), fluoroquinolones (ciprofloxacin, CIP5; nalidixic acid, NA30), sulfonamide combination (co-trimoxazole, SXT25), nitrofurantoin (F300), and tetracycline (TE30). *E. coli* ATCC25922 was used as the negative control. Isolates with intermediate or resistant phenotypes were classified as non-susceptible. Multidrug resistance was defined as non-susceptibility towards >3 antibiotic classes [38].

## Procurement of clinical ESBL-producing *Escherichia coli*

Hospital Segamat is the primary tertiary care provider in the Segamat district, where the community samples were collected [39]. ESBL-producing *E. coli* isolated from patients admitted from June through October 2020 were obtained from the pathology department. The study was approved by the Malaysian Medical Review and Ethics Committee (MREC, project ID NMRR-19-2532-50266) and MUHREC (project number 20722).

## Whole-genome sequencing of ESBL-producing *Escherichia coli*

A total of 40 ESBL-producing *Escherichia coli*, comprising 32 community and 8 clinical isolates, were further analyzed through whole-genome sequencing. The 32 community isolates were chosen based on their multidrug resistance profiles, while the eight clinical isolates were all the ESBL-EC isolated from Hospital Segamat during the sample collection period (June-October 2020). DNA was extracted using the QIAamp DNA Stool Mini Kit (Qiagen) and short-read sequenced using Illumina Miseq with a $2 \times 150$ bp paired-end configuration, giving $1,444,947 \pm 919,310$ mean raw read depths. All raw sequence data were trimmed to remove low-quality sequences and sequencing adapters using fastp version 0.20.1 [40], yielding a final count of $1,418,514 \pm 913,627$ mean reads post-trimming. BUSCO version 5.1.2 [41] was then run on the assembled sequence to confirm the completeness of orthologs from the sequence data (**S1 Fig**).

The isolates were sequence typed *in silico* using the Achtman scheme against the PubMLST database, conducted in SRST2 version 0.2.0 [42]. SRST2 was also used to determine the antibiotic resistance genes, plasmid replicon types, and virulence factors carried using the 'ARGannot_r3.fasta', 'plasmidFinder.fasta', and VFDB databases, respectively. The SRST2-curated databases are accessible through its repository at https://github.com/katholt/srst2/tree/master/data/. Chromosomal point mutations were identified using ResFinder version 4.0 [43]. ST131 subtype was analyzed using the ST131Typer version 1.0.0, available at https://github.com/JohnsonSingerLab/ST131Typer.

The antibiotic susceptibility, antibiotic resistance genes, multilocus sequence typing (MLST), and plasmid replicon profiles of the isolates were plotted and clustered using the hierarchical clustering method in the R package ComplexHeatmap version 2.4.3 [44]. Only MLST profiles with >1 count were included in the heatmap.

## Pan-genomic comparison of Segamat-derived ST131 isolates between settings and with regional variants

We further analyzed the presence of clonal transmission of ESBL-producing *Escherichia coli* in Segamat. The whole-genome sequences from each isolate were assembled using Unicycler

version 0.4.8 [45] and annotated using Prokka version 1.14.6 [46]. Pan-genome comparison was conducted using Roary version 3.13.0 [47]. The phylogenomic relationship was built using an approximately maximum likelihood tree using FastTree version 2.1.10 with the -gtr and -nt command [48], explored using Phandango version 1.3.0 [49], and visualized using the R package ggtree version 2.2.4 [50]. Parallelization of the pipeline utilized the GNU parallel platform [51].

Additionally, ST131 isolates (n = 5) were further analyzed through pan-genome comparison with publicly available ST131 sequences, focusing on those from the Southeast Asian region. We searched the Scopus database using the following search strategy: TITLE-ABS-KEY (("*Escherichia*\*" OR "*coli*\*") AND ("genome\*" OR "sequence\*") AND ("\*131\*") AND ("Malaysia\*" OR "Indonesia\*" OR "Singapore\*" OR "Thailand\*" OR "Vietnam\*" OR "Philippine\*" OR "Myanmar\*" OR "Burm\*" OR "Cambodia\*" OR "Lao\*" OR "Brunei\*" OR "Timor\*" OR "Chin\*" OR "Korea\*" OR "Japan\*")) AND (LIMIT-TO (AFFILCOUNTRY, "Thailand") OR LIMIT-TO(AFFILCOUNTRY, "Singapore") OR LIMIT-TO (AFFILCOUN-TRY, "Indonesia") OR LIMIT-TO (AFFILCOUNTRY, "Malaysia") OR LIMIT-TO (AFFIL-COUNTRY, "Cambodia") OR LIMIT-TO (AFFILCOUNTRY, "Myanmar") OR LIMIT-TO (AFFILCOUNTRY, "Viet Nam")). A total of 23 studies were filtered, out of which sequence data from four studies were eligible for further analysis (**S1** and **S2 Tables**). Out of the 670 procured sequences, 220 isolates were confirmed as ST131 using the sequence typing method described earlier.

## Statistical analyses

All statistical analyses were conducted in R version 4.0.5. Mixed model analysis was conducted to determine the factors significantly associated with ESBL carriage and the ESBL prevalence rate, adjusted for household clustering using the R package lme4 version 1.1–23 [52]. Ordination analysis was conducted using the R package vegan version 2.5–6 [53] and ape version 5.4–1 [54]. Correlation analyses were conducted using the R package corrplot version 0.90 [55]. Figures and plots were made using the R package ggplot2 version 3.3.3 [56].

## Results

### Community ESBL colonization was prevalent across demographics and comorbidities

A total of 233 fecal samples from 110 households in Segamat District in southern Malaysia were screened for the presence of ESBL-EC. The subjects were aged 43.65 ± SD 19.89 and were approximately equally distributed between sex (female, n = 127, 54.51%, $\chi^2$ test, p = 0.17) and different ethnicities ($\chi^2$ test, p = 0.45, **Table 1**). The most frequent occupations were home-makers (n = 64), agricultural workers (n = 43) and children (individuals <18 and not working, n = 33), with 42 subjects reporting unemployment.

A total of 15 participants reported having a surgery in the year prior to sampling, with dental surgery being the most frequent (n = 6, **S2A Fig**). The most frequent comorbidities were hypertension (n = 54), followed by high blood cholesterol (n = 28) and diabetes (n = 26) (**S2B Fig**). A total of 45 subjects were on active medication, most commonly with simvastatin (n = 17), amlodipine (n = 17), and metformin (n = 13) (**S2C Fig**).

Growth of cefotaxime-resistant *E. coli* was observed in isolates from 103 participants. Out of these, 44 participants were positive for ESBL-producing *E. coli* (ESBL-EC) based on the combination disk test. After accounting for the household clustering of the samples, this equates to a 17.82% (Linear mixed model, 95% CI: 10.48%– 24.11%) ESBL-EC carriage rate.

**Table 1. Demographic distribution of Segamat community dwellers (n = 233) recruited into this study.**

| Factor | Values | ESBL-positive | | ESBL-negative | | LRT |
|---|---|---|---|---|---|---|
| | | n | % | n | % | |
| Age | 10–25 | 15 | 0.34 | 43 | 0.23 | 0.13 |
| | 26–47 | 12 | 0.27 | 47 | 0.25 | |
| | 48–59 | 11 | 0.25 | 47 | 0.25 | |
| | 60–83 | 6 | 0.14 | 51 | 0.27 | |
| Sex | Female | 28 | 0.64 | 99 | 0.52 | 0.28 |
| | Male | 16 | 0.36 | 90 | 0.48 | |
| BMI | Underweight | 4 | 0.09 | 20 | 0.11 | 0.77 |
| | Normal | 16 | 0.36 | 83 | 0.45 | |
| | Overweight | 12 | 0.27 | 45 | 0.24 | |
| | Obese | 12 | 0.27 | 37 | 0.20 | |
| Ethnicity | Chinese | 11 | 0.25 | 58 | 0.31 | 0.48 |
| | Indian | 9 | 0.20 | 46 | 0.24 | |
| | Malay | 10 | 0.23 | 44 | 0.23 | |
| | Jakun | 14 | 0.32 | 41 | 0.22 | |
| Occupation | Agricultural | 11 | 0.26 | 32 | 0.17 | 0.48 |
| | Children | 7 | 0.16 | 26 | 0.14 | |
| | Homemaker | 13 | 0.30 | 51 | 0.27 | |
| | Others | 3 | 0.07 | 26 | 0.14 | |
| | Service | 4 | 0.09 | 17 | 0.09 | |
| | Unemployed | 5 | 0.12 | 37 | 0.20 | |
| Education | No formal education | 2 | 0.05 | 15 | 0.08 | 0.44 |
| | Did not complete primary school | 9 | 0.20 | 50 | 0.27 | |
| | Primary school | 10 | 0.23 | 45 | 0.24 | |
| | Penilaian Menengah Rendah (Lower Secondary Assessment) | 15 | 0.34 | 33 | 0.18 | |
| | Sijil Pelajaran Malaysia (Fifth form secondary school) | 6 | 0.14 | 35 | 0.19 | |
| | Diploma | 1 | 0.02 | 5 | 0.03 | |
| | Degree | 1 | 0.02 | 4 | 0.02 | |

Age was grouped based on quartiles. The occupation was classified based on the International Standard Classification of Occupations. Likelihood ratio test (LRT) p-value measures the significance of each factor to ESBL carriage. Blank data was removed from each variable before analysis.

Risk factor analysis did not reveal any significant association between any demographic factors with ESBL colonization (**Table 1**, Likelihood ratio test (LRT), p>0.05). The three most frequent comorbidities (hypertension, n = 54; cholesterol, n = 28; diabetes, n = 26) and surgical history were also not associated with ESBL (LRT, p>0.05). Similarly, the three most frequently used drugs (simvastatin, n = 17; amlodipine, n = 17; metformin, n = 13) and being on any active medications (n = 45) were also not significantly associated with ESBL colonization of the human gut (LRT, p>0.05).

Antibiotic susceptibility tests demonstrated the relatively higher and lower susceptibility of ESBL-EC towards ceftazidime (CAZ30) and cefepime (FEP30), respectively (**Fig 1**). A total of 90.29% (n = 93/103) of the cefotaxime-resistant *E. coli* were multidrug-resistant, with ESBL-EC being non-susceptible to significantly (Welch two Sample t-test, p<0.001) more antibiotics (mean 6.23 ± 1.10) compared to ESBL-negative isolates (5.29 ± 1.17). All of the tested isolates were susceptible to IPM10.

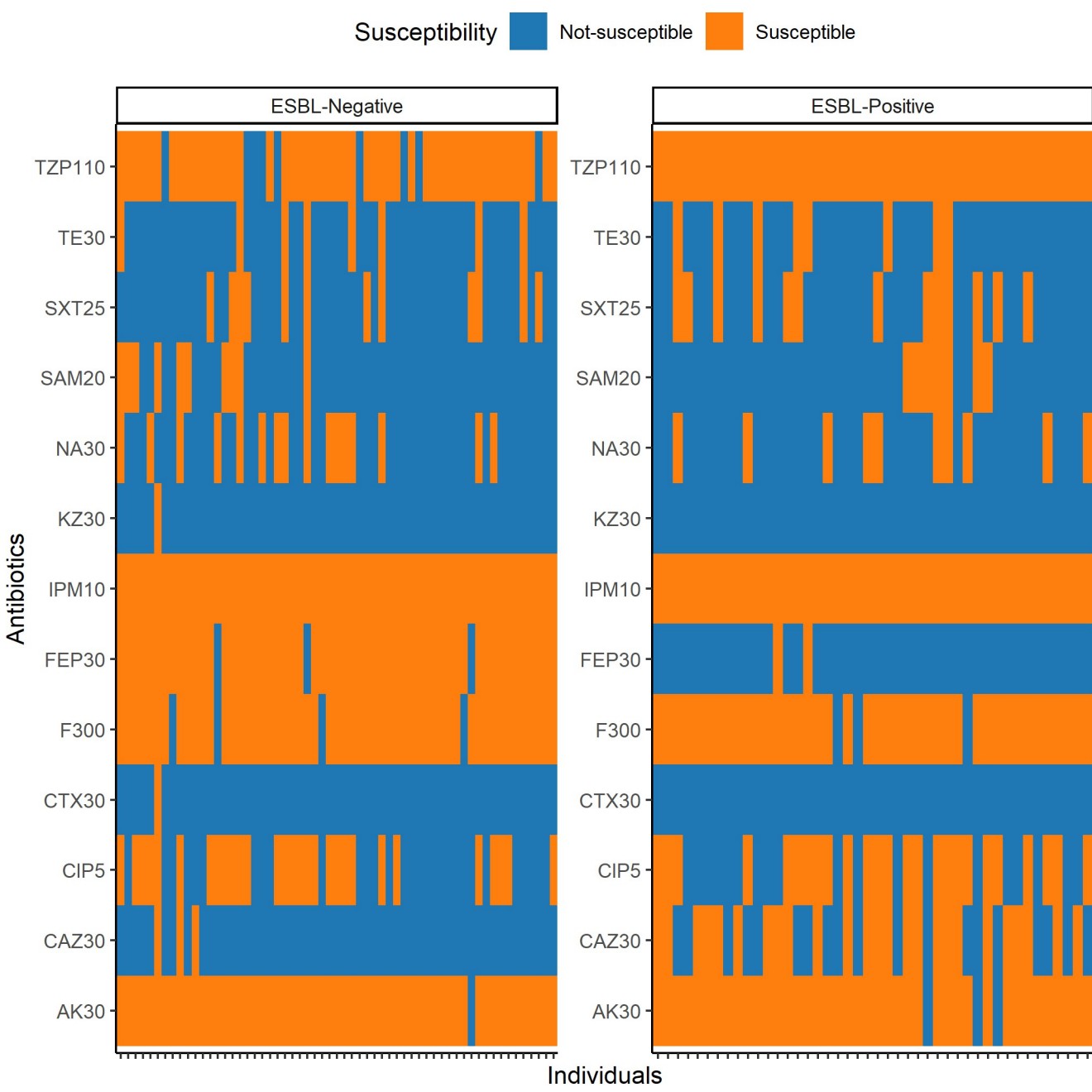

**Fig 1. Antibiotic susceptibility profile of independent *Escherichia coli* isolated from Segamat community dwellers against 13 antibiotics, sorted based on ESBL phenotype.** Abbreviations: TZP110 = Piperacillin-Tazobactam (110 μg); TE30 = Tetracycline (30 μg); SXT25 = Sulfamethoxazole-Trimethoprim (25 μg); SAM20 = Ampicillin-Sulbactam (20 μg); NA30 = Nalidixic Acid (30 μg); KZ30 = Cefazolin (30 μg); IPM10 = Imipenem (10 μg); FEP30 = Cefepime (30 μg); F300 = Nitrofurantoin (300 μg); CTX30 = Cefotaxime (30 μg); CIP5 = Ciprofloxacin (5 μg); CAZ30 = Ceftazidime (30 μg); AK30 = Amikacin (30 μg).

## Similar genotypic and phenotypic profiles across ESBL-EC isolated from the community and clinical setting

Out of the 44 ESBL-EC identified, 32 were selected based on their multidrug resistance profile for further analysis. Through whole-genome sequencing, we compared the genotypic and

**Table 2. Demographic profile and clinical characteristics of eight ESBL-producing *Escherichia coli* isolated from clinical patients admitted to the Segamat district hospital.**

| ID | Age | Sex | Source |
|----|-----|-----|--------|
| 40 | 53 | F | tracheal aspirate |
| 15 | 66 | M | NA |
| 16 | 55 | F | urine |
| 17 | 72 | F | blood |
| 18 | 58 | F | urine |
| 39 | 44 | F | urine |
| 14 | 61 | M | blood |
| 38 | 61 | M | NA |

phenotypic profiles of these isolates with eight clinical ESBL-EC isolated from independent patients admitted to the Segamat district hospital from June through October 2020 (mean age 58.75 ± 8.48, 62.5% female). These isolates were collected from various body sites, with urine (n = 3) and blood (n = 2) being the most common isolation source (**Table 2**).

Thirty-four unique resistance genes were detected from all the isolates, which expressed resistance to 11 antibiotic classes (**S3 Table**). The core genes *ampC* and *ampH* were universally carried by all *E. coli* isolates, but rarely conferred clinically relevant resistance [57]. Similarly, the universal carriage of *mrdA* reflected its role as an essential cell wall biosynthesis gene in *E. coli* [58]. Apart from this, *tetA* was the most frequently carried resistance gene (n = 31/40), followed by the $bla_{CTX-M-9}$ family (n = 22/40), which comprised $bla_{CTX-M-27}$ (n = 12) and $bla_{CTX-M-65}$ (n = 10). Of note, colistin resistance was detected in two isolates, each carrying the *mcr1* and *mcr3* gene, respectively. Among the PMQR genes, only *qnrS* was detected, which was carried by 37.5% (n = 15/40) of the isolates. No carbapenemase resistance genes were detected. Meanwhile, 28 unique plasmid replicon groups were detected, with each isolate carrying a mean of 4.45 ± 2.37 plasmid groups. FII was the most commonly encountered (n = 33/40), followed by FIB (n = 26) and I1 Alpha (n = 17). Additionally, a total of 335 virulence factors were observed (mean carriage 161.3 ± 26.21). Notably, no significant differences in the number of antibiotic resistance genes, plasmid replicon groups, and virulence factors carried were detected between the community and clinical isolates (Welch Two Sample t-test, p>0.05).

Analyzing the presence of five virulence gene markers for ExPEC strains: *papA*/*papC*, *afa*/*dra*, *sfa*/*foc*, *iutA*, and *kps* [59], five of eight (62.5%) clinical isolates were classified as ExPEC, while 28.1% (n = 9/32) of the community isolates were ExPEC (**S3 Fig**). Additionally, virulence gene markers for uropathogenic *E. coli* (UPEC) based on the presence of eight marker genes (*fyuA*, *yfcV*, *chuA*, *vat*, *focA*, *pap*, *sfa*, *cnf*) [60], enteroaggregative *E. coli* (detection of *aatA* and *aggR* [60]), and atypical enteropathogenic *E. coli* (EPEC-atypical, detected carriage of the *eae* gene [61]), were also frequently detected from the isolates, regardless of their setting. We further analyzed the 335 detected virulence factors to identify the top differentially abundant genes between the community and clinical isolates (**S4 Fig**). The clinical isolates had a higher carriage of *iutA*, *iuc*, *sit*, and *hly* virulence genes. Meanwhile, community isolates more frequently carried the *esp* gene. Despite these differences, 76.7% (n = 257/335) of these virulence genes were detected from isolates in both settings.

We analyzed whether antibiotic resistance gene carriage was associated with any resistance phenotypes based on antibiotic susceptibility profiles (**S5 Fig**). Phenotypic resistance towards CIP5 (ciprofloxacin) exhibited the most significant association (p<0.05) with antibiotic resistance genes (n = 9), followed by SXT25 (trimethoprim-sulfamethoxazole) (n = 7) and CAZ30

(ceftazidime) (n = 6). Notably, the correlation direction differed across antibiotic resistance genes belonging to the same groups. For example, carriage of aminoglycoside resistance genes *aadA* and *aadA4/5* were positively and negatively correlated with non-susceptibility towards ciprofloxacin, respectively. Additionally, $bla_{CTX-M-1}$ and $bla_{CTX-M-9}$ families were positively and negatively associated with CAZ30 non-susceptibility, respectively. Apart from this, several genotypic-phenotypic associations were consistent across antibiotic and resistance gene classes (e.g., *tetA* and TE30, *dfrA7* and SXT25).

Chromosomal point mutations in the QRDR genes were detected in 57.5% (n = 23/40) isolates. All these isolates had mutations in the *gyrA* gene, while 12 and 8 exhibited mutations in the *parC* and *parE* genes, respectively (S6 Fig). A total of 96.4% (n = 27/28) isolates showing non-susceptibility to fluoroquinolone possessed at least one PMQR or QRDR mutation. However, PMQR or QRDR mutations were also frequently detected among fluoroquinolone-susceptible isolates (75.0%, n = 9/12).

## Similar $bla_{CTX-M}$ distribution between the community and clinical isolates

$bla_{CTX-M-65}$ (n = 10) was the most frequently observed $bla_{CTX-M}$ variant from the community members, followed closely by $bla_{CTX-M-27}$ (n = 9) and $bla_{CTX-M-15}$ (n = 7) (Fig 2A). No signs of geographical clustering were observed, with all variants distributed throughout the study area (S7 Fig). Meanwhile, three $bla_{CTX-M}$ variants were observed among the clinical isolates, namely $bla_{CTX-M-55}$ (n = 3), $bla_{CTX-M-15}$ (n = 2), and $bla_{CTX-M-27}$ (n = 3). No isolates carried more than a single $bla_{CTX-M}$ gene.

A variety of strain types was observed in both settings, with ST131 (n = 3/8) and ST155 (n = 4/32) being the most frequently observed ST among the clinical and community isolates, respectively (Fig 2B). ST131 was also detected among the community isolates (n = 2). Further typing revealed that all clinical ST131 isolates belonged to Clade A with the O16:H5 serotype, while the community ST131 isolates were of the C1 clade with the serotype O25b:H4. The distinction between ST131 from the clinical and community settings was further confirmed through SNP-based pan-genome comparison with publicly available ST131 (n = 220), which revealed the segregation of the community and clinical isolates on different clades (Fig 3). Despite this, all isolates still clustered closely with clinical isolates from other geographical regions.

Within Segamat itself, SNP-based pan-genome comparison revealed the clustering of the isolates based on their MLST profile (Fig 4). Interestingly, some ST profiles are consistent with the types of $bla_{CTX-M}$ carried. For example, all ST131 isolates carried $bla_{CTX-M-27}$, while ST155 had the $bla_{CTX-M-65}$ gene. However, the carriage of $bla_{CTX-M}$ did not seem to drive the overall antibiotic resistance profiles of the isolates. This observation was revealed through a cluster analysis based on the antibiotic resistance genes and susceptibility profile, as well as plasmid groups (S4 Fig). Notably, four distinct clusters were observed, mainly distinguished by the types of aminoglycoside, phenicols, and fosfomycin resistance genes carried. For example, the aminoglycoside resistance genes *strA*, *strB* and *aadA4/5* were frequently co-carried with the trimethoprim resistance gene *dfrA7*. Meanwhile, the aminoglycoside resistance gene *aadA*, *aac3*, *aph4* and *aph3* were commonly co-carried with *dfrA*. Notably, the community and clinical isolates exhibited similar profiles and co-clustered together. Ordination of the isolates using the Jaccard distance further confirmed this observation (Fig 5).

## Discussion

This study investigated the prevalence of fecal colonization with ESBL-EC from 233 community dwellers in the Segamat district, and compared their profiles with clinical ESBL-EC

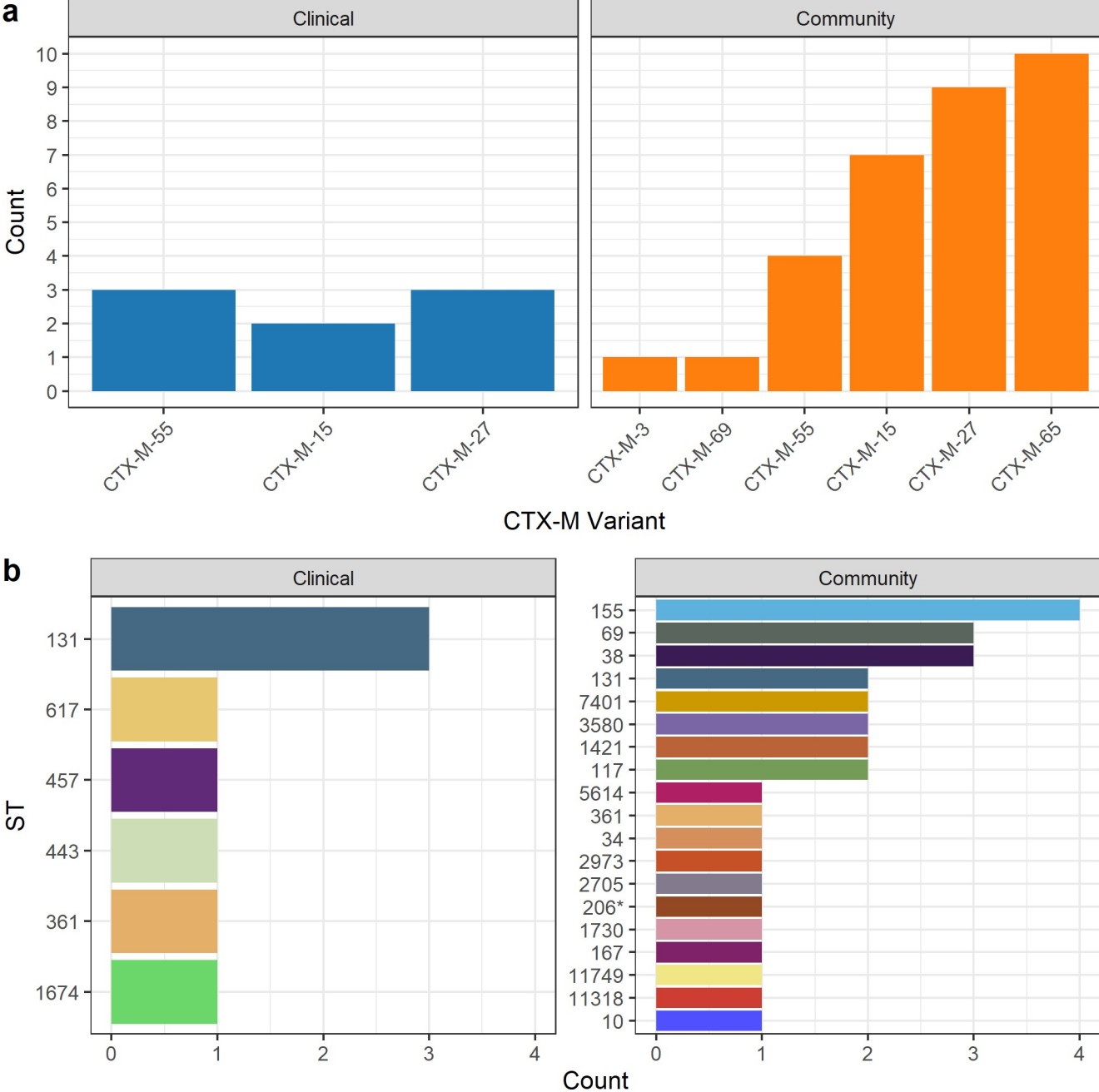

**Fig 2.** Distribution of $bla_{CTX-M}$ variant (a) and MLST (b) carried by ESBL-producing *E. coli* isolated from Segamat clinical and community samples. MLST was typed using the Achtman scheme.

isolated from patients in the same district. We report for the first time the association between ESBL-EC isolated from the asymptomatic community and a co-located healthcare setting in Malaysia. Isolates from both settings shared similar resistance genes, susceptibility profiles and carried plasmid groups, suggesting that horizontal gene transfer is a dominant dissemination route for ESBL and other antibiotic resistance genes in the region. Additionally, we believe this to be the first report of ESBL colonization among community dwellers in Malaysia.

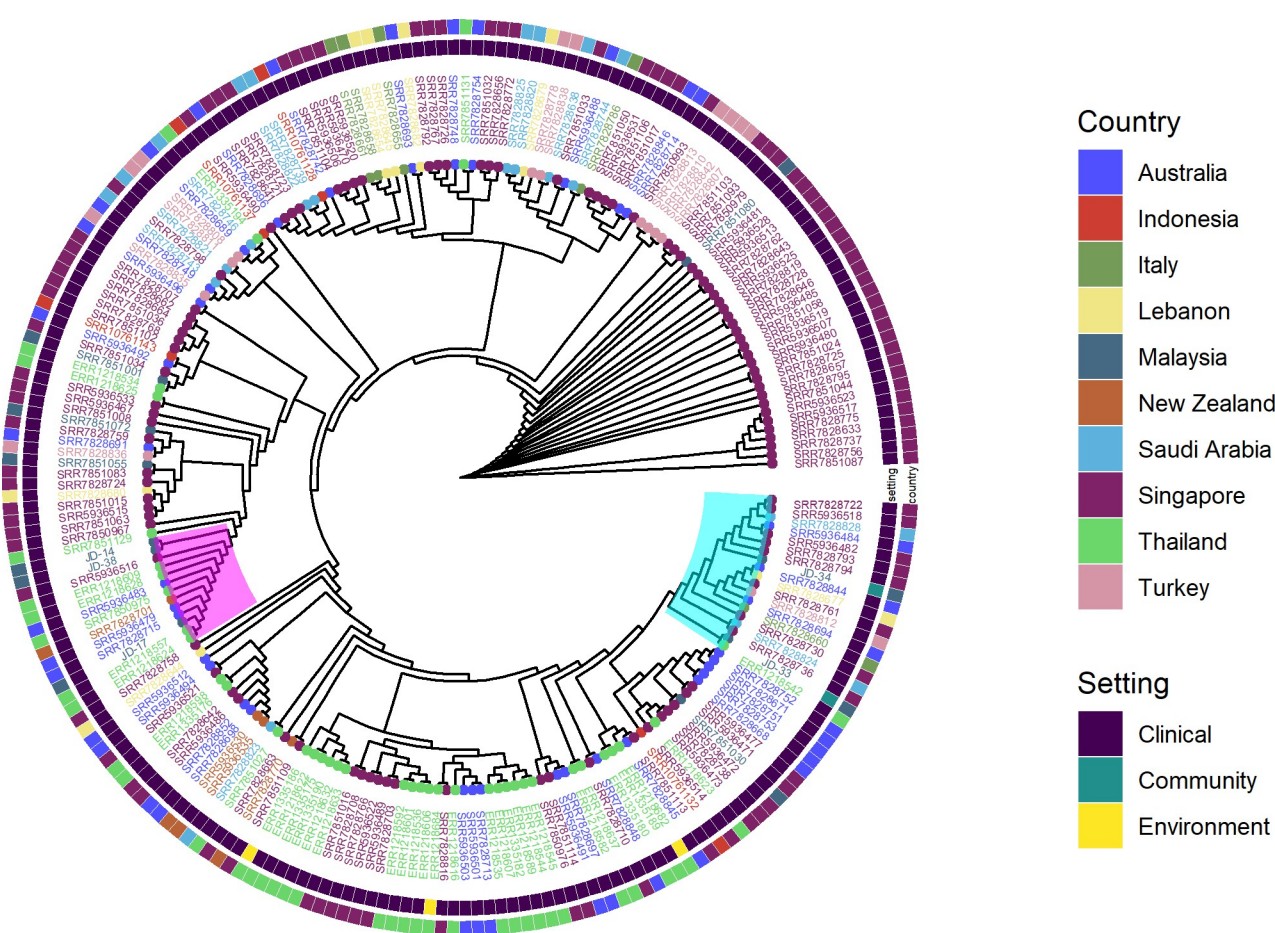

**Fig 3. SNP-based pan-genome comparison of ESBL-producing *Escherichia coli* ST 131 from Segamat (n = 5) with public ST131 sequences (n = 220).** The clade containing the community ST131 isolates from Segamat was highlighted in turquoise, while the Segamat clinical ST131 isolates were mapped in the clade highlighted in purple.

The 17.82% prevalence rate of ESBL-EC among community dwellers was similar to the global average of 16.5% reported in a recent meta-analysis [24]. However, it was lower than the South-East Asia mean of 27%, but still almost three-fold higher than Europe [24]. Within Southeast Asia, the prevalence rate reported in this study for Malaysia was even lower than neighbouring countries: Singapore (26.2%) [3] and Thailand (52.1%) [2].

Age has been reported to be a risk factor for ESBL-associated infections [62, 63]. Additionally, ESBL colonization has also been associated with demographic factors such as ethnicity [64] and education level [65]. New Zealanders of South Asian descent were more likely to travel to South Asia, a hotbed for $bla_{CTX-M-15}$ [64]). Better-educated individuals in China were more susceptible to ESBL colonization, likely due to a higher likelihood of consuming antibiotics [65]. Our failure to associate ESBL colonization with age and other demographic parameters indicates the endemicity of ESBL in the community. The lack of association between ESBL and other comorbidities, as well as surgical history, also reinforced this observation. This situation might have resulted from the lack of antibiotic regulation enforcement in the region [66]. A previous report has postulated the lack of antibiotic regulation enforcement as a factor driving similar ESBL profiles between animal and clinical isolates in Malaysia [35]. Moreover, plasmids carrying ESBL have been reported to be persistent and could be stably inherited despite

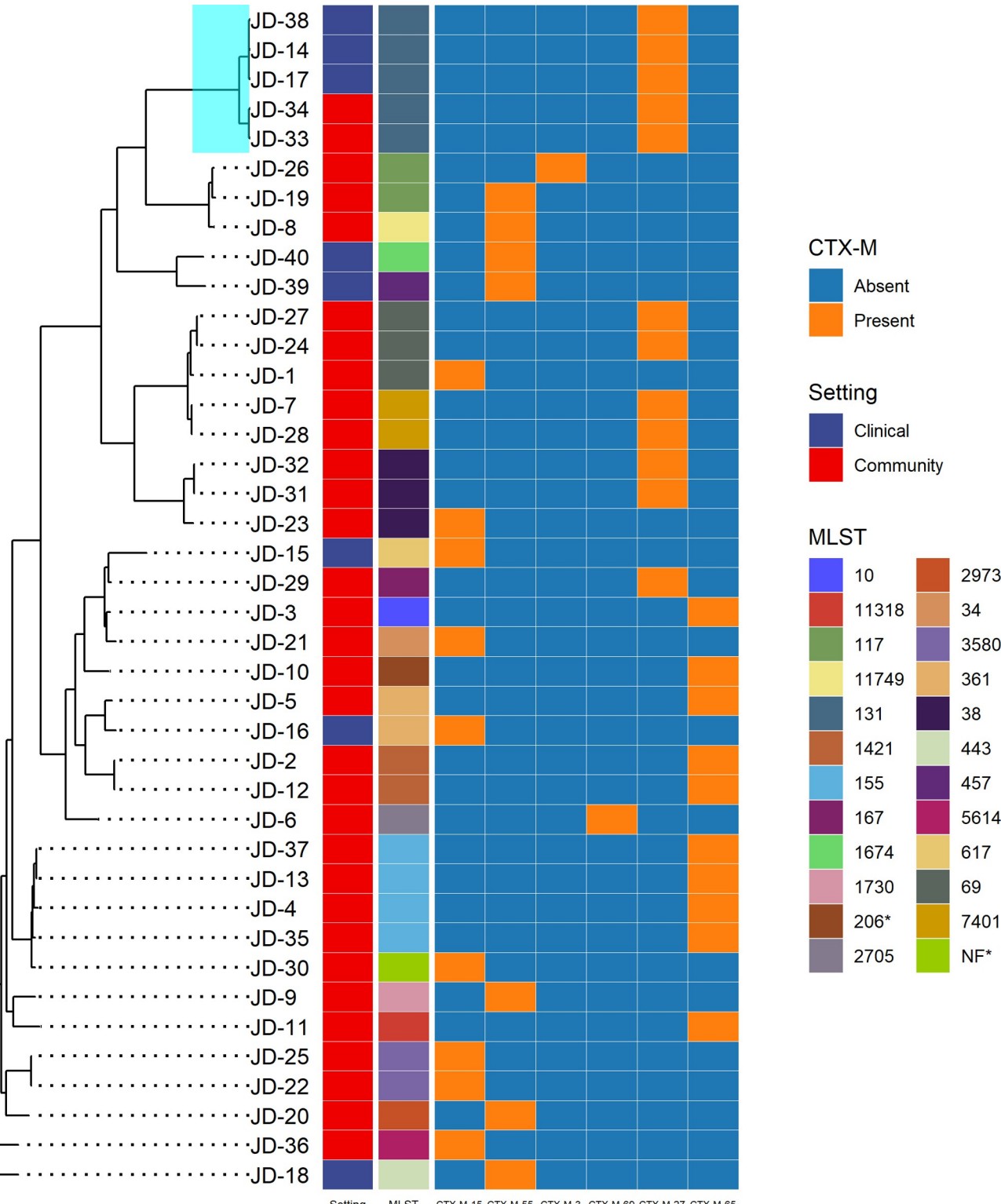

**Fig 4. SNP-based pan-genome comparison of ESBL-producing *Escherichia coli* isolated from Segamat community members (n = 32) and hospital patients (n = 8), annotated with their setting, MLST, and *bla*$_\text{CTX-M}$ variant carried.** The ST131 clade is highlighted in turquoise.

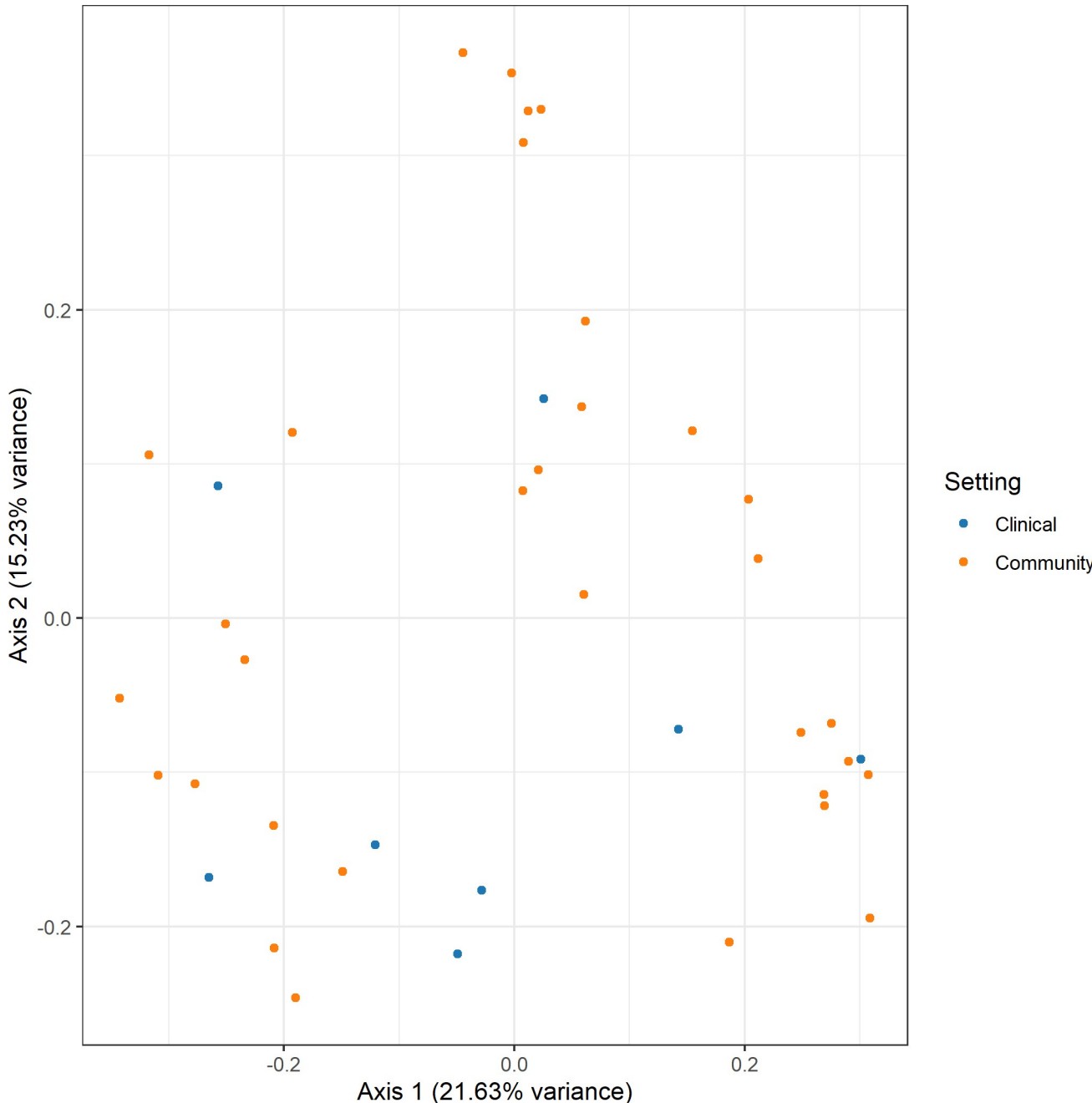

**Fig 5. Principal coordinate analysis with Jaccard distance based on the isolates' antibiotic resistance genes, susceptibility profiles, and carried plasmid groups.**

the absence of antibiotic selection pressure [21, 67]. The high prevalence of ESBL in the community might reflect the successful propagation of plasmids carrying ESBL and other antibiotic resistance genes introduced from past events, reinforced by the lack of antibiotic regulation in the region.

A high proportion of the clinical isolates carried virulence genes that have been linked to ExPEC strains. The *iutA* gene encodes for the aerobactin receptor, established as a marker

gene for ExPEC [59]. Additionally, the frequent detection of *iucABCD*, which encodes for aerobactin and has been associated with Avian Pathogenic *E. coli* (APEC) [68] and UPEC [69], suggests their role in driving the pathogenicity of the clinical isolates in Segamat. The frequent detection of *hlyCABD* operons, encoding α-hemolysins, was also in line with reports suggesting their association with UPEC [70]. Meanwhile, the community isolates frequently carried Type III secretion system effector-like protein (*espL4*, *espX4*, *espX5*, *espX1*, *espR1*), associated with Shiga-Toxin producing *E. coli* (STEC) [71] and the enterohemorrhagic *E. coli* (EHEC) [61]. However, the Shiga toxin gene itself was absent from the isolates. This observation was confirmed by the frequent classification of isolates from both the community and clinical settings into various pathogroups, suggesting their ability to readily cause infections, highlighting the importance of proper antibiotic surveillance and control on the asymptomatic community.

A variety of STs was detected from the community members, reflecting the vast diversity of bacterial strains which populate the gastrointestinal tract [72, 73]. In contrast, clinical isolates were often dominated by a few pathogenic and hypervirulent ST, indicating the occurrence of a clinical outbreak [74–76]. The absence of a clonal relationship between the community and clinical isolates in our study is expected due to the different nature and isolation sources of the isolates. The fecally derived community isolates were commensal in nature compared to the clinical pathogenic isolates which were procured from extraintestinal infection sites. Nevertheless, the observed similarity in the resistome profiles of both the commensal community isolates and the pathogenic clinical isolates suggests the frequent exchange of genetic materials between isolates of both settings. This exchange can occur when contamination occurs through the fecal-oral route, resulting in the transmission of ESBL-producing isolates between individuals [19, 77] and can potentially lead to the exchange of genetic materials between commensal and pathogenic strains, highlighting concerns on the role of commensal isolates in the gastrointestinal tract as a reservoir for ESBL and other antibiotic resistance genes. Regardless, this study is not equipped to unveil the directionality of this association, warranting further study.

Although ST131 isolates were observed from both the community and clinical settings, they belonged to different clades and serotypes. Despite this, ESBL-EC isolates from the community and clinical settings frequently shared similar antibiotic susceptibility, plasmid, and resistance genes profiles. Our observation suggests the long-term stability and persistence of the mobile genetic elements carrying ESBL and other antibiotic resistance determinants in the region.

All ST131 in our study carried the $bla_{\text{CTX-M-27}}$, similar to the dominant ST131 clone reported in Japan. However, this result is incongruent with recent findings by Chen et al. [11], who observed the dominance of ST131 SEA-C2 clade associated with $bla_{\text{CTX-M-15}}$ in Southeast Asian isolates causing bacteremia. Nevertheless, the low sample size observed in our cohort was inconclusive.

The non-susceptibility of isolates carrying $bla_{\text{CTX-M-9}}$ to ceftazidime has been reported [78], likely explaining the non-susceptibility of isolates carrying the $bla_{\text{CTX-M-9}}$ family gene towards ceftazidime. Additionally, all tested isolates only carried a single $bla_{\text{CTX-M}}$ gene, indicating the carriage of $bla_{\text{CTX-M-1}}$ and $bla_{\text{CTX-M-9}}$ on different plasmid groups belonging to the same incompatibility groups. Their carriage on plasmids of the same incompatibility group likely explains the absence of co-carriage of both $bla_{\text{CTX-M}}$ genes in Segamat.

Despite the frequent detection of aminoglycoside resistance genes, the susceptibility rates of the tested ESBL-EC against amikacin remains high. The observed susceptibility towards amikacin is consistent with the literature, where ESBL-producing isolates from Malaysia are generally susceptible towards amikacin (e.g., 94.6% [79], 98% [80] and 100% [32] susceptibility rate). Additionally, the resistance of amikacin against most aminoglycoside-modifying genes is commonly reported (as reviewed in [81]). Notably, amikacin resistance is reported to be

mediated by aminoglycoside genes such as *aphA6* [82], *armA* [83], *aacA4* and *aacA7* [84], none of which were detected in the Segamat cohort.

Resistance towards fluoroquinolone antibiotics can be mediated by PMQR and QRDR [85, 86]. Fluoroquinolone resistance was highly variable among ESBL producers in Malaysia, ranging from 18% to 71% [32, 80, 87]. Previously, a study on ciprofloxacin-resistant *K. pneumoniae* identified the *gyrA* and *parC* QRDR mutations as the driver of fluoroquinolone resistance in Malaysia [79]. This observation was also accurate for our cohort, with *gyrA* and *parC* chromosomal mutations frequently detected. The PMQR gene *qnrS* was also frequently detected from the tested ESBL isolates. The combination of PMQR and QRDR genes seemed to drive fluoroquinolone resistance in Segamat, although a large proportion of fluoroquinolone-susceptible isolates also carried at least one PMQR/QRDR mutation. This observation might have implied the lack of PMQR expression, as reported before. Regardless of their susceptibility, fluoroquinolone-susceptible isolates carrying a single *qnr* gene have been demonstrated to rapidly gain fluoroquinolone resistance upon challenge with fluoroquinolone antibiotics [88], presenting a concern despite their susceptibility.

Carbapenem resistance is an emerging global concern partly due to its increased frequency of usage to treat ESBL-related infections [89], including in the Southeast Asian region [90]. Although carbapenem resistance was not detected from the Segamat cohort, this might be related to the low positivity rate of carbapenem resistance, ranging from 3.5–4.1% in Malaysia [90]. Nevertheless, the absence of carbapenem resistance in Segamat confirms the preservation of carbapenem's efficacy as the last line antibiotic in Segamat. Despite this, a long-term study is warranted to gauge the emergence of carbapenem resistance in Segamat.

The different resistome clusters observed in this study seem to be driven by the aminoglycoside and phenicol resistance genes. This observation implies multiple co-carriage of different resistance genes, in particular those conferring resistance to aminoglycosides and phenicols, in plasmids of the same incompatibility groups. The multidrug resistance observed concurred with previous findings linking the ESBL-carrying plasmids with co-carriage of other resistance genes [91], although the type $bla_{CTX-M}$ carried in our cohort did not explain the cluster differences. The lack of dual carriage of $bla_{CTX-M}$ likely indicates that $bla_{CTX-M}$ variants were carried in plasmids in identical incompatibility groups, warranting further investigations into the plasmid-$bla_{CTX-M}$ relationship in Segamat. Additionally, the detection of *mcr3* warrants further research on their potential impact on colistin resistance in Segamat considering its role as a last-line antibiotic [92]. Moreover, the observed correlation between IncHI2 and IncHI2A with *mcr3* warrants further plasmid-gene investigation. Although we did not study the resistome profile of the non-ESBL isolates, it is worth noting that even ESBL-negative isolates had a mean non-susceptibility towards 5.29 ± 1.17 antibiotic classes, suggesting the endemicity of antibiotic resistance beyond ESBL in the Southeast Asian region.

Malaysia is a prime destination for international tourists, recording more than 26 million tourist arrivals in 2019, worth MYR86.14 billion in tourist receipts [93]. Additionally, 239.1 million domestic tourists were recorded in the year 2019 [94]. These data suggest the risk of ESBL dissemination both within and beyond the Southeast Asian region, potentially aggravating the ongoing global ESBL crisis once travel resumes after easing of restrictions imposed by the current COVID-19 pandemic.

This study is not free from limitations. The relatively low overall sample size might have hampered the detection of demographic risk factors of ESBL colonization. Additionally, due to the ethics requirements, we did not obtain information on patient admission dates, which hampered the classification of the clinical isolates into either hospital or community-acquired. Nevertheless, these limitations do not void the observed resistome similarity between the community and clinical isolates.

Our study compared fecal ESBL-EC from the community with clinical ESBL-EC derived from various extraintestinal environments. The inherent limitation of this design was the non-comparability of the mostly commensal community isolates with the pathogenic clinical isolates, limiting information on the clonal relatedness of isolates from both settings. However, our study design was able to directly associate the high ESBL colonization rate in the community with ESBL-associated infections in the clinical setting.

Our observation was limited to the human samples (i.e. clinical and the community). As such, we were unable to elucidate the presence of ESBL in settings such as food and animal farms. Additionally, the reliance on only short-read sequence data means that plasmid assembly data were unavailable, which hindered the elucidation of the relationship between the different $bla_{CTX-M}$ variants and plasmid types. *In silico* plasmid analysis of short-read sequences (e.g. plasmidSpades) was ineffective given the low copy number of plasmids carrying ESBL [95]. The high frequency of plasmid types and resistance genes detected also complicated the pairwise correlation analysis.

Our study design did not account for the presence of ESBL genes which might have been phenotypically masked due to the overexpression of *AmpC*, which is not susceptible to the ß-lactam/ß-lactam inhibitor combination [96]. As a result, the prevalence rate of ESBL colonization reported in this study might be lower than the actual colonization rate of ESBL in Segamat.

Additionally, the cross-sectional nature of our study was insufficient to account for the resistome dynamics both in the community and clinical settings. Moreover, the lack of temporal sampling further hampers our ability to track the transmission dynamics of ESBL in the region. Despite these caveats, our result still provides a robust analysis of the community resistome profiles and hints at community-clinical transmission of ESBL.

We profiled the prevalence of ESBL-EC in the community and hospital settings in southern Malaysia. We observed similar profiles between the community and clinical isolates, based on the types of plasmids, antibiotic resistance genes, and virulence factors carried, implying the frequent exchange of genetic materials through horizontal gene transfer between the two settings. Despite a one-year sampling gap between the community and clinical isolates, the similarity in profiles suggests the persistence and stable inheritance of these antibiotic resistance determinants. A comprehensive multi-year carriage study is warranted to capture the temporal trend of transmission and devise better public health measures to curb the transmission of antibiotic resistance.

## Supporting information

**S1 Fig. BUSCO assessment of the ESBL-producing *Escherichia coli* isolates (n = 40) subjected to whole-genome sequencing.**
(PDF)

**S2 Fig.** Health demographic data of the subjects included in the study, including (a) surgical history in the past year, (b) comorbidities, (c) and active medication.
(PDF)

**S3 Fig. Pathogroup classification of ESBL-producing *E. coli* based on the presence of virulence gene markers.**
(PDF)

**S4 Fig. Top 40 virulence genes with the biggest proportional difference between the community and clinical isolates.**
(PDF)

**S5 Fig. Correlation analysis of antibiotic susceptibility profiles of ESBL-producing *E. coli* and detected antibiotic resistance genes based on SRST2.**
(PDF)

**S6 Fig. Phenotypic and genotypic profiles of ESBL-producing *E. coli* based on fluoroquinolone susceptibility profiles of the isolates and the presence of PMQR gene or QRDR mutations.** Abbreviations: CIP5 = Ciprofloxacin (5 μg); NA30 = Nalidixic Acid (30 μg).
(PDF)

**S7 Fig. Geographical isolation site of ESBL-producing *E. coli* from Segamat community dwellers annotated with their *bla*$_{CTX-M}$ variant status.**
(PDF)

**S8 Fig. Clustered heatmap of ESBL-producing *Escherichia coli* isolated from Segamat community members (n = 32) and hospital patients (n = 8) based on the presence of antibiotic resistance genes, plasmid replicon types, MLST, and phenotypic susceptibility profiles.** Community isolates are labelled in blue, while clinical isolates are labelled in red. Abbreviations: Fcn = fosfomycin; Phe = phenicols; MLS = macrolides and lincosamides; Tet = tetracycline; Agl = aminoglycoside; Tmt = trimethoprim; Col = colistin; Flq = fluoroquinolone; AST = Antibiotic susceptibility profile.
(PDF)

**S1 Table. List of studies with publicly available ST131 short read sequence data included for pan-genome analysis.**
(CSV)

**S2 Table. Details of ST131 isolates included in the pan-genome analysis.**
(CSV)

**S3 Table. List of detected genes from ESBL-producing *Escherichia coli* in Segamat through whole-genome sequencing.**
(CSV)

## Acknowledgments

The authors are grateful to the SEACO field team and Dr. Faidzal Adlee Bin Abdul Ghafar for their support in the data collection and isolate procurement used in this study.

## Author Contributions

**Conceptualization:** Jacky Dwiyanto, Daniel Reidpath, Shaun Wen Huey Lee, Qasim Ayub, Sui Mae Lee, Sadequr Rahman.

**Formal analysis:** Jacky Dwiyanto, Chun Wie Chong.

**Funding acquisition:** Sui Mae Lee, Sadequr Rahman.

**Investigation:** Jacky Dwiyanto, Jia Wei Hor, Tin Tin Su, Su Chern Foo, Sadequr Rahman.

**Methodology:** Jacky Dwiyanto, Daniel Reidpath, Shaun Wen Huey Lee, Qasim Ayub, Sui Mae Lee, Chun Wie Chong, Sadequr Rahman.

**Resources:** Faizah Binti Mustapha, Sadequr Rahman.

**Supervision:** Daniel Reidpath, Qasim Ayub, Sui Mae Lee, Su Chern Foo, Chun Wie Chong, Sadequr Rahman.

**Writing – original draft:** Jacky Dwiyanto, Su Chern Foo, Chun Wie Chong, Sadequr Rahman.

**Writing – review & editing:** Jacky Dwiyanto, Jia Wei Hor, Daniel Reidpath, Tin Tin Su, Shaun Wen Huey Lee, Qasim Ayub, Faizah Binti Mustapha, Sui Mae Lee, Su Chern Foo, Chun Wie Chong, Sadequr Rahman.

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
