## [Decision Letter · Decision Letter 0]

8 Oct 2021

PONE-D-21-27273Pan-genome and resistome analysis of extended-spectrum ß-lactamase-producing Escherichia coli: a multi-setting epidemiological surveillance study from MalaysiaPLOS ONE

Dear Dr. Dwiyanto,

Thank you for submitting your manuscript to PLOS ONE. After careful consideration, we feel that it has merit but does not fully meet PLOS ONE’s publication criteria as it currently stands. Therefore, we invite you to submit a revised version of the manuscript that addresses the points raised during the review process.

Please address all comments and questions raised by the reviewers. 

We look forward to receiving your revised manuscript.

Kind regards,

Iddya Karunasagar

Academic Editor

PLOS ONE

Journal Requirements:

Additional Editor Comments (if provided):

The reviewers have raised a number of questions and asked for several clarifications. Please address reviewer comments point by point.

Reviewers' comments:

Reviewer's Responses to Questions

**Comments to the Author**

1. Is the manuscript technically sound, and do the data support the conclusions?

Reviewer #1: Yes

Reviewer #2: Yes

Reviewer #3: Yes

2. Has the statistical analysis been performed appropriately and rigorously? 

Reviewer #1: Yes

Reviewer #2: Yes

Reviewer #3: Yes

3. Have the authors made all data underlying the findings in their manuscript fully available?

Reviewer #1: Yes

Reviewer #2: Yes

Reviewer #3: No

4. Is the manuscript presented in an intelligible fashion and written in standard English?

Reviewer #1: Yes

Reviewer #2: Yes

Reviewer #3: Yes

5. Review Comments to the Author

Reviewer #1: Pan-genome and resistome analysis of extended-spectrum ß-lactamase-producing Escherichia coli: a multi-setting epidemiological surveillance study from Malaysia

In this study, ESBL-producing Escherichia coli (ESBL-EC) were isolated from faecal samples, tested for antibiotic susceptibilities, and selected ESBL-EC were whole genome sequenced. Based on WGS analysis, plasmid replicon groups, resistance genes and virulence genogroups were identified. Genome comparison of community isolates with the clinical , isolates from the same region did not reveal clonal relatedness. However, the community and clinical isolates could be clustered into 4 groups based on the similarity with respect to antibiotic susceptibility and plasmid profiles, and the resistance genes carried.

Comments:

1. Although the study suggests no clonal relationship between the clinical and community isolates, this is based on a proportion of isolates subjected to WGS from diverse sources. The major drawback of this study is the comparison of isolates from unrelated sources; faecal isolates from community and extraintestinal isolates from the hospitals. This might explain the lack of clonality among the isolates used in this study.

2. The community isolates were from faecal samples, while the clinical isolates were from different sources It is obvious that the E. coli associated with GI tract and those found in extraintestinal niches could be clonally different, and phenotypically, these could belong to different pathogroups. Ideally, the comparisons should have been between faecal isolates from the community and the clinical sources

3. In lines 347-350, the authors state that their observation on the lack of clonality among clinical and community isolates was inconclusive due to one year gap in sampling between the two. However, they have emphasized on the absence of clonal relationship as one of the major findings of the study throughout the manuscript.

4. It is surprising to note that only blaCTX variants could be detected among the sequenced isolates. However, the distribution of ESBL genes in other isolates of this study is not known. The authors should have PCR screened all the isolates for major ESBL genes.

5. Nothing has been said about fluoroquinolone and carbapenem resistance among the isolates.

6. The study does not make attempt to compare the antibiotic resistance phenotypes with the genotypes. With the WGS available for a sizeable number of isolates, it is worth making this comparison.

7. The resistance genes against classes of antibiotics identified in the plasmids from WGS should be listed. The WGS analysis should have included QRDRs and the PMQRs

8. Line 252: Was there any association between the ESBL phenotype, and the virulence characteristics of clinical and community isolates? Can these isolates be groped into specific E. coli pathogroups based on these? (Like EPEC, EHEC etc).

9. How many isolates were recovered from each faecal sample?

10. What were the criteria used for selecting community and clinical isolates for whole genome sequencing? This should be clearly explained in the methods section. Importantly, the authors did not explain why they chose all or did they have any criteria for selecting isolates for phylogenetic analysis.

11. Table 1 uses demographic factors such as the ethnicity, occupation and education, while similar data is available for clinical samples. I could not determine from the manuscript how these factors, particularly occupation and education, have influence on faecal carriage of ESBL E. coli. Even if they have, the number of individuals sampled is too small to arrive at a definite conclusion.

12. L204: How many colonies were selected for analysis from each of these samples? Since no enrichment has been done here, it is presumed that the colonies that come up on cefotaxime plates could be, by far, non-clonal.

13. The study screened 233 fecal samples from 110 households, and the growth of cefotaxime-resistant E. coli was observed from 103 participants. However, ESBL-EC were detected in isolates from 44 participants. Nearly 60% of those isolates that grew on cefotaxime plate did not produce ESBL or the ESBL phenotype was not detectable by combination disc method. While inducible AmpC could be responsible for this, the absence of ESBLs in these should have been confirmed by PCR.

14. Line 364: Were these isolates colistin resistant? What was the MIC?

15. Abstract says 32 isolates were whole genome sequences. Please correct it as 40.

Reviewer #2: This paper explores the extent and pattern of the penetrance of ESBL-producing E.coli in a community versus clinical setting in Segamat, Malaysia. The paper is based on the hypothesis that communities could unknowingly harbor unregulated sources of ESBL-producing E. coli. Hypervirulent strains and horizontal plasmid transfer are potential mechanisms for the penetrance of this antimicrobial resistance from clinical settings into the community. Understanding the relationship between clinical antimicrobial resistance and community resistance through whole-genome sequencing could lead to better surveillance and spread of antimicrobial resistance outside of clinical settings.

In this paper, the authors used phenotypic profiling of ESBL-producing E.coli followed by whole-genome sequencing to compare the phenotypic and genotypic profiles of the bacteria in the clinical versus community setting. They compared CTX-M variants and SNPs pan-genome between the settings, which seems to be a robust analysis for understanding the similarities between the bacteria found in each of these settings.

Interestingly, the authors found that there were similar susceptibility profiles and plasmid groups between the ESBL-producing E.coli found in the clinical and community settings. This implies that there is a connection between the ESBL-producing E.coli colonization found in these two groups.

The strengths of this paper include that it provides the first community profile of ESBL-producing E.coli in Malaysia, which is essential for understanding the scope of the problem, and for targeting interventions. Another strength is the conduction of phenotypic profiling alongside the pan-genome analysis, which increases the fidelity of the results rather than stand-alone genotypic profiling. A potential criticism is that, as mentioned in the paper, the limitation of temporal sampling of decreases the scope to which the penetrance of ESBL-producing E.coli in the community setting can be understood from this paper. However, that could be a topic of future research and does not take away from the important community profile this paper provides.

Reviewer #3: The manuscript aimed describes the prevalence of extended-spectrum ß-lactamase-producing

Escherichia coli (ESBL-EC) associated with faecal samples from the community dwellers of Segamat town in Malaysia and compared their resistome and genomic profiles with ESBL-EC isolated from hospital patients in the same district, although with a sample gap of one year. Pan genome comparison and cluster analysis revealed four distinct clusters with similar resistome profiles between that the community isolates and clinical isolates thus suggesting the horizontal exchange of genetic material between the isolates.

The manuscript is well written and the authors have rightly described the few limitations of the study also highlighting the need to carry out similar studies in future.

After thoroughly reading the article, I would like to make the following comments and suggestions.

1] Introduction – Briefly highlight the importance of sequence type -ST131 in the introduction section. Line 85-87- Objectives need to be more clearly defined.

2] Materials and Methods-

a. Line 181- BioProject PRJNA752611 is not accessible on NCBI website.

b. Line 183- github.com/jdwiyanto/esbl_segamat is not accessible on website

3] Results-

a. Table 1- Education- PMR and SPM- Give full forms please

b. Table 2- Date of isolation of samples and diagnosis need not be mentioned. What was the source of Sample ID 15 and 38?

c. Table 2- Justify why isolates from faecal samples were not used for comparison with community isolates which were of faecal origin?

d. Line 289-291- Names of genes should be written in lower case italics. Please follow the nomenclatures as per journal guidelines

e. Supplementary Data- S4_figure, the nomenclature for isolates is different than that used through the rest of the manuscript. Please keep it uniform.

4] Discussion – Discussion is well written; however, it need not have subheadings.

5] References- Please correct the citation style as per journal requirements.

Final Remark: Overall, in my opinion, the manuscript provides significant information about the prevalence and spread of ESBL-EC in Southeast Asian countries, the work is well organised, all ethical approvals have been taken and the data presented is clear to understand. I would therefore suggest that the manuscript needs minor corrections.

6. PLOS authors have the option to publish the peer review history of their article (what does this mean?). If published, this will include your full peer review and any attached files.

Reviewer #1: No

Reviewer #2: No

Reviewer #3: No

---

## [Author Response · Author response to Decision Letter 0]

15 Dec 2021

The manuscript has been revised to conform to the journal’s formatting requirement.

The accession number to the raw data used in this manuscript has been provided in the manuscript (PRJNA752611). 

Additional Editor Comments (if provided):

The reviewers have raised a number of questions and asked for several clarifications. Please address reviewer comments point by point.

Reviewer #1: Pan-genome and resistome analysis of extended-spectrum ß-lactamase-producing Escherichia coli: a multi-setting epidemiological surveillance study from Malaysia

In this study, ESBL-producing Escherichia coli (ESBL-EC) were isolated from faecal samples, tested for antibiotic susceptibilities, and selected ESBL-EC were whole genome sequenced. Based on WGS analysis, plasmid replicon groups, resistance genes and virulence genogroups were identified. Genome comparison of community isolates with the clinical , isolates from the same region did not reveal clonal relatedness. However, the community and clinical isolates could be clustered into 4 groups based on the similarity with respect to antibiotic susceptibility and plasmid profiles, and the resistance genes carried.

Comments:

1. Although the study suggests no clonal relationship between the clinical and community isolates, this is based on a proportion of isolates subjected to WGS from diverse sources. The major drawback of this study is the comparison of isolates from unrelated sources; faecal isolates from community and extraintestinal isolates from the hospitals. This might explain the lack of clonality among the isolates used in this study.

We thank the Reviewer for highlighting this point. The primary purpose of the study design in comparing the commensal community isolates derived from feces with clinical isolates of extraintestinal origins was to determine whether the community could be linked to ESBL-associated extraintestinal infections affecting the clinical setting. Although fecal-to-fecal comparison might unveil a clonal relationship between the community and clinical isolates, this information would not inform whether the high colonization rate of ESBL in Segamat was associated with clinical infections. We thank the Reviewer for highlighting the lack of clarity this manuscript was trying to convey and have revised the introduction (line 88-92) and discussion sections (line 409-414) of the manuscript accordingly. We have also included discussion on the non-equal sample comparison in line 403-406. 

Lines 88-92: Despite the known endemicity of ESBLs in the Southeast Asian community, there is a lack of comparative genomic analysis of community and clinical isolates, resulting in a gap in our understanding of the relationship between the commensal ESBL-producing isolates in the community and those causing extraintestinal infections in the region. Unveiling such a link is necessary to inform proper surveillance and antibiotic regulation policies to curb the further spread of ESBL in the region.

Lines 403-406: The absence of a clonal relationship between the community and clinical isolates in our study is expected due to the different nature and isolation sources of the isolates. The fecally derived community isolates were commensal in nature compared to the clinical pathogenic isolates which were procured from extraintestinal infection sites.

Lines 409-414: This exchange can occur when contamination occurs through the fecal-oral route, resulting in the transmission of ESBL-producing isolates between individuals [79, 80] and can potentially lead to the exchange of genetic materials between commensal and pathogenic strains, highlighting concerns on the role of commensal isolates in the gastrointestinal tract as a reservoir for ESBL and other antibiotic resistance genes.

2. The community isolates were from faecal samples, while the clinical isolates were from different sources It is obvious that the E. coli associated with GI tract and those found in extraintestinal niches could be clonally different, and phenotypically, these could belong to different pathogroups. Ideally, the comparisons should have been between faecal isolates from the community and the clinical sources

This comment is related to comment #1. Although a direct comparison with fecal samples procured from the clinical patients might yield a clonal relationship between the community and clinical isolates, it would not inform the impact of the high colonization rate of ESBL in the region on the clinical settings. We have added a statement on the rationale behind the commensal – extraintestinal isolates comparison in line 403-406.

3. In lines 347-350, the authors state that their observation on the lack of clonality among clinical and community isolates was inconclusive due to one year gap in sampling between the two. However, they have emphasized on the absence of clonal relationship as one of the major findings of the study throughout the manuscript.

We thank the Reviewer for highlighting the ambiguity of the message we conveyed in the original draft. The statement in question was meant to emphasize the similarity in the resistome despite the isolates being clonally unrelated. We have revised the aim of this study (line 88-92) and toned down statements highlighting the clonal unrelatedness of the isolates (lines 361, 416-417), as the primary message of the manuscript was on the similarity of the resistome profiles of commensal isolates with those causing extraintestinal infections regardless of their clonality. As mentioned in comment #2, we have also addressed the study design comparing commensal and extraintestinal isolates in lines 403-406.

Lines 88-92: Despite the known endemicity of ESBLs in the Southeast Asian community, there is a lack of comparative genomic analysis of community and clinical isolates, resulting in a gap in our understanding of the relationship between the commensal ESBL-producing isolates in the community and those causing extraintestinal infections in the region.

Line 361: Isolates from both settings shared similar resistance genes, susceptibility profiles and carried plasmid groups despite being clonally unrelated, suggesting that horizontal gene transfer is a dominant dissemination route for ESBL and other antibiotic resistance genes in the region.

Lines 416-417: Although ST131 isolates were observed from both the community and clinical settings, they belonged to different clades and serotypes. , suggesting the lack of clonal transmission between the two settings in Segamat

4. It is surprising to note that only blaCTX variants could be detected among the sequenced isolates. However, the distribution of ESBL genes in other isolates of this study is not known. The authors should have PCR screened all the isolates for major ESBL genes.

We agree with the Reviewer that screening for all isolates would yield a more accurate distribution of ESBL and other resistance genes in the region. Our screening criteria on the phenotypic expression of ESBL might have missed several ESBL variants which have a masked phenotypic ESBL expression (such as OXA), as well as those expressing AmpC ß-lactamase. We have added this as a limitation of this study in lines 504-508.

Lines 504-508: Our study design did not account for the presence of ESBL genes which might have been phenotypically masked due to the overexpression of AmpC, which is not susceptible to the ß-lactam/ß-lactam inhibitor combination [99]. As a result, the prevalence rate of ESBL colonization reported in this study might be lower than the actual colonization rate of ESBL in Segamat.

5. Nothing has been said about fluoroquinolone and carbapenem resistance among the isolates.

We have included a new analysis on fluoroquinolone resistance in Results section lines 271-272 and lines 304-309, which describes the presence of PMQR and QRDR. We have also briefly added PMQR and QRDR on the introduction lines 71-74 and discussed them in lines 441-454. Discussion on carbapenem resistance has been added to the manuscript (lines 455-461).

Lines 71-74: Additionally, ST131 is also frequently associated with fluoroquinolone resistance, either through the carriage of plasmid-mediated quinolone resistance (PMQR) genes such as qnrS or quinolone resistance determining region (QRDR) chromosomal mutations, such as gyrA and parC [14, 17].

Lines 271-272: Among the PMQR genes, only qnrS was detected, which was carried by 37.5% (n=15/40) of the isolates. 

Lines 304-309: Chromosomal point mutations in the QRDR genes were detected in 57.5% (n=23/40) isolates. All these isolates had mutations in the gyrA gene, while 12 and 8 exhibited mutations in the parC and parE genes, respectively (S6 Fig). A total of 96.4% (n=27/28) isolates showing non-susceptibility to fluoroquinolone possessed at least one PMQR or QRDR mutation. However, PMQR or QRDR mutations were also frequently detected among fluoroquinolone-susceptible isolates (75.0%, n=9/12). 

Lines 441-454: Resistance towards fluoroquinolone antibiotics can be mediated by PMQR and QRDR [88, 89]. Fluoroquinolone resistance was highly variable among ESBL producers in Malaysia, ranging from 18% to 71% [33, 83, 90]. Previously, a study on ciprofloxacin-resistant K. pneumoniae identified the gyrA and parC QRDR mutations as the driver of fluoroquinolone resistance in Malaysia [82]. This observation was also accurate for our cohort, with gyrA and parC chromosomal mutations frequently detected. The PMQR gene qnrS was also frequently detected from the tested ESBL isolates. The combination of PMQR and QRDR genes seemed to drive fluoroquinolone resistance in Segamat, although a large proportion of fluoroquinolone-susceptible isolates also carried at least one PMQR/QRDR mutation. This observation might have implied the lack of PMQR expression, as reported before . Regardless of their susceptibility, fluoroquinolone-susceptible isolates carrying a single qnr gene have been demonstrated to rapidly gain fluoroquinolone resistance upon challenge with fluoroquinolone antibiotics [91], presenting a concern despite their susceptibility.

Lines 455-461: Carbapenem resistance is an emerging global concern partly due to its increased frequency of usage to treat ESBL-related infections [92], including in the Southeast Asian region [93]. Although carbapenem resistance was not detected from the Segamat cohort, this might be related to the low positivity rate of carbapenem resistance, ranging from 3.5-4.1% in Malaysia [93]. Nevertheless, the absence of carbapenem resistance in Segamat confirms the preservation of carbapenem’s efficacy as the last line antibiotic in Segamat. Despite this, a long-term study is warranted to gauge the emergence of carbapenem resistance in Segamat.

6. The study does not make attempt to compare the antibiotic resistance phenotypes with the genotypes. With the WGS available for a sizeable number of isolates, it is worth making this comparison.

We have added a new analysis to correlate antibiotic susceptibility phenotype with the detected antibiotic resistance genes in lines 293-303. A new figure denoting this analysis has also been added as S5 Fig. In the discussion section, we further discussed the association between phenotypic susceptibility and gene presence in lines 427-454. The methods section has also been revised to describe the correlation analysis conducted (lines 198-199).

Lines 198-199: Correlation analyses were conducted using the R package corrplot version 0.90 [56].

Lines 293-303: We analyzed whether antibiotic resistance gene carriage was associated with any resistance phenotypes based on antibiotic susceptibility profiles (S5 Fig). Phenotypic resistance towards CIP5 (ciprofloxacin) exhibited the most significant association (p<0.05) with antibiotic resistance genes (n=9), followed by SXT25 (trimethoprim-sulfamethoxazole) (n=7) and CAZ30 (ceftazidime) (n=6). Notably, the correlation direction differed across antibiotic resistance genes belonging to the same groups. For example, carriage of aminoglycoside resistance genes aadA and aadA4/5 were positively and negatively correlated with non-susceptibility towards ciprofloxacin, respectively. Additionally, blaCTX-M-1 and blaCTX-M-9 families were positively and negatively associated with CAZ30 non-susceptibility, respectively. Apart from this, several genotypic-phenotypic associations were consistent across antibiotic and resistance gene classes (e.g., tetA and TE30, dfrA7 and SXT25).

Lines 427-454: The non-susceptibility of isolates carrying blaCTX-M-9 to ceftazidime has been reported [81], likely explaining the non-susceptibility of isolates carrying the blaCTX-M-9 family gene towards ceftazidime. Additionally, all tested isolates only carried a single blaCTX-M gene, indicating the carriage of bla¬CTX-M-1 and blaCTX-M-9 on different plasmid groups belonging to the same incompatibility groups. Their carriage on plasmids of the same incompatibility group likely explains the absence of co-carriage of both blaCTX-M genes in Segamat.

Despite the frequent detection of aminoglycoside resistance genes, the susceptibility rates of the tested ESBL-EC against amikacin remains high. The observed susceptibility towards amikacin is consistent with the literature, where ESBL-producing isolates from Malaysia are generally susceptible towards amikacin (e.g., 94.6% [82], 98% [83] and 100% [33] susceptibility rate). Additionally, the resistance of amikacin against most aminoglycoside-modifying genes is commonly reported (as reviewed in [84]). Notably, amikacin resistance is reported to be mediated by aminoglycoside genes such as aphA6 [85], armA [86], aacA4 and aacA7 [87], none of which were detected in the Segamat cohort. 

Resistance towards fluoroquinolone antibiotics can be mediated by PMQR and QRDR [88, 89]. Fluoroquinolone resistance was highly variable among ESBL producers in Malaysia, ranging from 18% to 71% [33, 83, 90]. Previously, a study on ciprofloxacin-resistant K. pneumoniae identified the gyrA and parC QRDR mutations as the driver of fluoroquinolone resistance in Malaysia [82]. This observation was also accurate for our cohort, with gyrA and parC chromosomal mutations frequently detected. The PMQR gene qnrS was also frequently detected from the tested ESBL isolates. The combination of PMQR and QRDR genes seemed to drive fluoroquinolone resistance in Segamat, although a large proportion of fluoroquinolone-susceptible isolates also carried at least one PMQR/QRDR mutation. This observation might have implied the lack of PMQR expression, as reported before . Regardless of their susceptibility, fluoroquinolone-susceptible isolates carrying a single qnr gene have been demonstrated to rapidly gain fluoroquinolone resistance upon challenge with fluoroquinolone antibiotics [91], presenting a concern despite their susceptibility.

7. The resistance genes against classes of antibiotics identified in the plasmids from WGS should be listed. The WGS analysis should have included QRDRs and the PMQRs

We did not specify resistance genes exclusively from plasmids in our study due to the lack of a complete genome assembly for our isolates. The use of PlasmidFinder to analyze plasmids carrying ESBL and its co-carried resistance genes is not ideal due to the typically low copy number of plasmids carrying ESBL, resulting in an underdetection (lines 498-501). However, we uploaded all the resistance genes, plasmids, and virulence factors identified through the whole genome sequences of the isolates as S3 Table. 

The query on PMQR and QRDR is related to comment #5. We have added new analyses on PMQR and QRDR in lines 271-272 and 304-309 and added a brief introduction (lines 71-74) and discussion (lines 441-454) on them. 

Lines 498-501: Additionally, the reliance on only short-read sequence data means that plasmid assembly data were unavailable, which hindered the elucidation of the relationship between the different blaCTX-M variants and plasmid types. In silico plasmid analysis of short-read sequences (e.g. plasmidSpades) was ineffective given the low copy number of plasmids carrying ESBL [98].

8. Line 252: Was there any association between the ESBL phenotype, and the virulence characteristics of clinical and community isolates? Can these isolates be groped into specific E. coli pathogroups based on these? (Like EPEC, EHEC etc).

We have added a new analysis that identified virulence markers associated with E. coli pathogroups from both the community and clinical isolates in results lines 280-292. This analysis is now discussed in lines 386-399.

Lines 280-292: Analyzing the presence of five virulence gene markers for ExpEC strains: papA/papC, afa/dra, sfa/foc, iutA, and kps [60], five of eight (62.5%) clinical isolates were classified as ExPEC, while 28.1% (n=9/32) of the community isolates were ExPEC (S3 Fig). Additionally, virulence gene markers for uropathogenic E. coli (UPEC) based on the presence of eight marker genes (fyuA, yfcV, chuA, vat, focA, pap, sfa, cnf) [61], enteroaggregative E. coli (detection of aatA and aggR [61]), and atypical enteropathogenic E. coli (EPEC-atypical, detected carriage of the eae gene [62]), were also frequently detected from the isolates, regardless of their setting. We further analyzed the 335 detected virulence factors to identify the top differentially abundant genes between the community and clinical isolates (S4 Fig). The clinical isolates had a higher carriage of iutA, iuc, sit, and hly virulence genes. Meanwhile, community isolates more frequently carried the esp gene. Despite these differences, 76.7% (n=257/335) of these virulence genes were detected from isolates in both settings.

Lines 386-399: A high proportion of the clinical isolates carried virulence genes that have been linked to ExPEC strains. The iutA gene encodes for the aerobactin receptor, established as a marker gene for ExPEC [60]. Additionally, the frequent detection of iucABCD, which encodes for aerobactin and has been associated with Avian Pathogenic E. coli (APEC) [69] and UPEC [70], suggests their role in driving the pathogenicity of the clinical isolates in Segamat. The frequent detection of hlyCABD operons, encoding α-hemolysins, was also in line with reports suggesting their association with UPEC [71]. Meanwhile, the community isolates frequently carried Type III secretion system effector-like protein (espL4, espX4, espX5, espX1, espR1), associated with Shiga-Toxin producing E. coli (STEC) [72] and the enterohemorrhagic E. coli (EHEC) [73]. However, the Shiga toxin gene itself was absent from the isolates. This observation was confirmed by the frequent classification of isolates from both the community and clinical settings into various pathogroups, suggesting their ability to readily cause infections, highlighting the importance of proper antibiotic surveillance and control on the asymptomatic community.

9. How many isolates were recovered from each faecal sample?

One isolate with Escherichia coli morphology was randomly isolated from each fecal sample (line 118-119).

Lines 118-119: From each sample, one presumptive E. coli isolate was randomly picked using Harrison's disk method.

10. What were the criteria used for selecting community and clinical isolates for whole genome sequencing? This should be clearly explained in the methods section. Importantly, the authors did not explain why they chose all or did they have any criteria for selecting isolates for phylogenetic analysis.

We thank the Reviewer for pointing out the lack of clarity on the selection criteria in the original draft. The 32 community isolates were chosen based on their multidrug resistance profiles and is now described in lines 147-148. Additionally, the eight clinical isolates were all the ESBL-producing E. coli which was isolated from the hospital during the sample collection period (June-October 2020) and is described in lines 148-150. The relatively short hospital sample collection period was due to the lockdown imposed by the Malaysian government, hampering the originally scheduled collection period of one year (September 2019 – October 2020). 

Lines 147-148: The 32 community isolates were chosen based on their multidrug resistance profiles, 

Lines 148-150: while the eight clinical isolates were all the ESBL-EC isolated from Hospital Segamat during the sample collection period (June-October 2020).

11. Table 1 uses demographic factors such as the ethnicity, occupation and education, while similar data is available for clinical samples. I could not determine from the manuscript how these factors, particularly occupation and education, have influence on faecal carriage of ESBL E. coli. Even if they have, the number of individuals sampled is too small to arrive at a definite conclusion.

The demographic information was obtained to determine whether any specific population group were at a higher risk of ESBL colonization. Although demographic factors such as age, ethnicity, and education have been linked with ESBL colonization, it has, to the best of our knowledge, not been tested in a region with a very high ESBL colonization prevalence, such as Southeast Asia. We observed no significant association with any demographic factors tested, suggesting the endemicity of ESBL colonization in Segamat. We have described this in lines 370-385. However, we agree with the Reviewer’s concern that our relatively low sample size might have been unable to uncover these associations and have added this as a study limitation in lines 484-485.

Lines 370-385: Age has been reported to be a risk factor for ESBL-associated infections [63, 64]. Additionally, ESBL colonization has also been associated with demographic factors such as ethnicity [65] and education level [66]. New Zealanders of South Asian descent were more likely to travel to South Asia, a hotbed for blaCTX-M-15 [65]). Better-educated individuals in China were more susceptible to ESBL colonization, likely due to a higher likelihood of consuming antibiotics [66]. Our failure to associate ESBL colonization with age and other demographic parameters indicates the endemicity of ESBL in the community. The lack of association between ESBL and other comorbidities, as well as surgical history, also reinforced this observation. This situation might have resulted from the lack of antibiotic regulation enforcement in the region [67]. A previous report has postulated the lack of antibiotic regulation enforcement as a factor driving similar ESBL profiles between animal and clinical isolates in Malaysia [36]. Moreover, plasmids carrying ESBL have been reported to be persistent and could be stably inherited despite the absence of antibiotic selection pressure [22, 68]. The high prevalence of ESBL in the community might reflect the successful propagation of plasmids carrying ESBL and other antibiotic resistance genes introduced from past events, reinforced by the lack of antibiotic regulation in the region.

Lines 484-485: The relatively low overall sample size might have hampered the detection of demographic risk factors of ESBL colonization.

12. L204: How many colonies were selected for analysis from each of these samples? Since no enrichment has been done here, it is presumed that the colonies that come up on cefotaxime plates could be, by far, non-clonal.

One colony with E. coli morphology growing on the cefotaxime plate was isolated for further analysis. Each fecal sample obtained was enriched in 1:10 buffered peptone water prior to culture on the cefotaxime plate (lines 116-119). 

Lines 116-119: Within 24 h of expulsion, each fecal sample was diluted 1:10 in buffered peptone water (Oxoid) and spread plated on MacConkey agar (Oxoid) laced with two mg/L cefotaxime (Gold Biotechnology), and then incubated overnight at 37°C. From each sample, one presumptive E. coli isolate was randomly picked using Harrison's disk method.

13. The study screened 233 fecal samples from 110 households, and the growth of cefotaxime-resistant E. coli was observed from 103 participants. However, ESBL-EC were detected in isolates from 44 participants. Nearly 60% of those isolates that grew on cefotaxime plate did not produce ESBL or the ESBL phenotype was not detectable by combination disc method. While inducible AmpC could be responsible for this, the absence of ESBLs in these should have been confirmed by PCR.

We agree with the Reviewer’s concern that some ESBL-producing E. coli might have been phenotypically masked by the presence of inducible AmpC, hence underreporting the colonization rate of ESBL-producing E. coli in Segamat. We have highlighted this as a study limitation in lines 504-508.

Lines 504-508: Our study design did not account for the presence of ESBL genes which might have been phenotypically masked due to the overexpression of AmpC, which is not susceptible to the ß-lactam/ß-lactam inhibitor combination [99]. As a result, the prevalence rate of ESBL colonization reported in this study might be lower than the actual colonization rate of ESBL in Segamat.

14. Line 364: Were these isolates colistin resistant? What was the MIC?

Unfortunately, the MIC of colistin sulfate and polymyxin B was not tested for these isolates, which we highlight as an aspect for future study in lines 470-472. 

Lines 470-472: Additionally, the detection of mcr3 warrants further research on their potential impact on colistin resistance in Segamat considering its role as a last-line antibiotic [95].

15. Abstract says 32 isolates were whole genome sequences. Please correct it as 40.

Thank you for pointing this out. We have revised the abstract accordingly in lines 33-34.

Lines 33-34: Whole-genome sequencing was then conducted on selected ESBL-EC from both settings (n=40) for pan-genome comparison, cluster analysis, and resistome profiling.

Reviewer #2: This paper explores the extent and pattern of the penetrance of ESBL-producing E.coli in a community versus clinical setting in Segamat, Malaysia. The paper is based on the hypothesis that communities could unknowingly harbor unregulated sources of ESBL-producing E. coli. Hypervirulent strains and horizontal plasmid transfer are potential mechanisms for the penetrance of this antimicrobial resistance from clinical settings into the community. Understanding the relationship between clinical antimicrobial resistance and community resistance through whole-genome sequencing could lead to better surveillance and spread of antimicrobial resistance outside of clinical settings.

In this paper, the authors used phenotypic profiling of ESBL-producing E.coli followed by whole-genome sequencing to compare the phenotypic and genotypic profiles of the bacteria in the clinical versus community setting. They compared CTX-M variants and SNPs pan-genome between the settings, which seems to be a robust analysis for understanding the similarities between the bacteria found in each of these settings.

Interestingly, the authors found that there were similar susceptibility profiles and plasmid groups between the ESBL-producing E.coli found in the clinical and community settings. This implies that there is a connection between the ESBL-producing E.coli colonization found in these two groups.

The strengths of this paper include that it provides the first community profile of ESBL-producing E.coli in Malaysia, which is essential for understanding the scope of the problem, and for targeting interventions. Another strength is the conduction of phenotypic profiling alongside the pan-genome analysis, which increases the fidelity of the results rather than stand-alone genotypic profiling. A potential criticism is that, as mentioned in the paper, the limitation of temporal sampling of decreases the scope to which the penetrance of ESBL-producing E.coli in the community setting can be understood from this paper. However, that could be a topic of future research and does not take away from the important community profile this paper provides.

We thank the Reviewer for the positive outlook on our work.

Reviewer #3: The manuscript aimed describes the prevalence of extended-spectrum ß-lactamase-producing

Escherichia coli (ESBL-EC) associated with faecal samples from the community dwellers of Segamat town in Malaysia and compared their resistome and genomic profiles with ESBL-EC isolated from hospital patients in the same district, although with a sample gap of one year. Pan genome comparison and cluster analysis revealed four distinct clusters with similar resistome profiles between that the community isolates and clinical isolates thus suggesting the horizontal exchange of genetic material between the isolates.

The manuscript is well written and the authors have rightly described the few limitations of the study also highlighting the need to carry out similar studies in future.

After thoroughly reading the article, I would like to make the following comments and suggestions.

We are thankful for the positive outlook of Reviewer #3 on our manuscript. 

1] Introduction – Briefly highlight the importance of sequence type -ST131 in the introduction section. Line 85-87- Objectives need to be more clearly defined.

We have added more descriptions on ST131 in the introduction section lines 63-77. Additionally, we have also elaborated the objectives of our study to clarify the central question the manuscript is addressing in lines 99-104. 

Lines 63-77: The successful propagation of ESBL genes has been linked to the hypervirulent strain Escherichia coli ST131 [11]. Since its emergence in the late 2000s [12, 13], E. coli ST131 gradually became a major strain causing extraintestinal infections worldwide (e.g., the dominance of ST131 among isolates causing bacteremia in Southeast Asia [11]). Its rapid emergence is driven by the successful acquisition of various virulence factors associated with extraintestinal pathogenic E. coli (ExPEC), such as the iutA aerobactin receptor and papG P fimbrial adhesin virulence genes [14]. Its role in disseminating ESBL lies in its frequent carriage of plasmid groups carrying the blaCTX-M gene, which is frequently co-carried with other resistance genes, particularly aminoglycosides [15, 16]. Additionally, ST131 is also frequently associated with fluoroquinolone resistance, either through the carriage of plasmid-mediated quinolone resistance (PMQR) genes such as qnrS or quinolone resistance determining region (QRDR) chromosomal mutations, such as gyrA and parC [14, 17]. Nevertheless, ESBL dissemination can also be carried and disseminated by commensal strains through horizontal gene transfer of plasmids carrying the ESBL gene [18], as observed in community studies (e.g., [7, 19]).

Lines 99-104: We aimed to address this gap by determining the colonization rate of ESBL-EC from community dwellers in Malaysia. Fecal samples obtained from a community cohort in Segamat, Malaysia, were screened for ESBL-EC. Clinical ESBL-EC isolates from the local hospital were also procured. Afterwards, the community and clinical isolates were compared based on their genome and resistomes through whole-genome sequencing. 

2] Materials and Methods-

a. Line 181- BioProject PRJNA752611 is not accessible on NCBI website.

BioProject PRJNA752611 has been released and is now publicly available. 

b. Line 183- github.com/jdwiyanto/esbl_segamat is not accessible on website

We have made the GitHub link to the analysis and pipeline script publicly available.

3] Results-

a. Table 1- Education- PMR and SPM- Give full forms please

We thank the Reviewer for highlighting this minor fault. The complete forms of PMR and SPM have been given in Table 1 (line 218).

Table 1, line 218: 

Penilaian Menengah Rendah (Lower Secondary Assessment)

Sijil Pelajaran Malaysia (Fifth form secondary school)

b. Table 2- Date of isolation of samples and diagnosis need not be mentioned. What was the source of Sample ID 15 and 38?

The date of isolation and diagnosis have been removed from Table 2 (line 263). Unfortunately, the hospital administration did not record the source of isolate ID 15 and 38.

c. Table 2- Justify why isolates from faecal samples were not used for comparison with community isolates which were of faecal origin?

The comparison between the commensal fecal isolates with the clinical isolates procured from extraintestinal origin was designed to detect whether the ESBL isolates carried by the community members can be associated with those causing infections (lines 405-409). We have addressed this in the Discussion section lines 490-495.

Lines 405-409: The fecally derived community isolates were commensal in nature compared to the clinical pathogenic isolates which were procured from extraintestinal infection sites. Nevertheless, the observed similarity in the resistome profiles of both the commensal community isolates and the pathogenic clinical isolates suggests the frequent exchange of genetic materials between isolates of both settings.

Lines 490-495: Our study compared fecal ESBL-EC from the community with clinical ESBL-EC derived from various extraintestinal environments. The inherent limitation of this design was the non-comparability of the mostly commensal community isolates with the pathogenic clinical isolates, limiting information on the clonal relatedness of isolates from both settings. However, our study design was able to directly associate the high ESBL colonization rate in the community with ESBL-associated infections in the clinical setting.

d. Line 289-291- Names of genes should be written in lower case italics. Please follow the nomenclatures as per journal guidelines

Thank you for highlighting this error. We have revised all gene names to lowercase italics as per the journal guideline. 

e. Supplementary Data- S4_figure, the nomenclature for isolates is different than that used through the rest of the manuscript. Please keep it uniform.

Thank you for pointing out this discrepancy. We have revised the ID labelling of the figure (now S8 Fig) to be consistent with the rest of the manuscript. 

4] Discussion – Discussion is well written; however, it need not have subheadings.

Subheadings have been removed from the discussion section

5] References- Please correct the citation style as per journal requirements.

The citations in the manuscript have been revised as per the journal guideline.

Final Remark: Overall, in my opinion, the manuscript provides significant information about the prevalence and spread of ESBL-EC in Southeast Asian countries, the work is well organised, all ethical approvals have been taken and the data presented is clear to understand. I would therefore suggest that the manuscript needs minor corrections.

We are grateful for the constructive feedbacks provided by Reviewer #3 and have revised the manuscript accordingly.

---

## [Decision Letter · Decision Letter 1]

24 Feb 2022

Pan-genome and resistome analysis of extended-spectrum ß-lactamase-producing Escherichia coli: a multi-setting epidemiological surveillance study from Malaysia

PONE-D-21-27273R1

Dear Dr. Dwiyanto,

We’re pleased to inform you that your manuscript has been judged scientifically suitable for publication and will be formally accepted for publication once it meets all outstanding technical requirements.

Kind regards,

Iddya Karunasagar

Academic Editor

PLOS ONE

Additional Editor Comments (optional):

All reviewer comments have been addressed satisfactorily.

Reviewers' comments:

Reviewer's Responses to Questions

**Comments to the Author**

1. If the authors have adequately addressed your comments raised in a previous round of review and you feel that this manuscript is now acceptable for publication, you may indicate that here to bypass the “Comments to the Author” section, enter your conflict of interest statement in the “Confidential to Editor” section, and submit your "Accept" recommendation.

Reviewer #1: (No Response)

Reviewer #3: All comments have been addressed

2. Is the manuscript technically sound, and do the data support the conclusions?

Reviewer #1: (No Response)

Reviewer #3: Yes

3. Has the statistical analysis been performed appropriately and rigorously? 

Reviewer #1: (No Response)

Reviewer #3: Yes

4. Have the authors made all data underlying the findings in their manuscript fully available?

Reviewer #1: (No Response)

Reviewer #3: Yes

5. Is the manuscript presented in an intelligible fashion and written in standard English?

Reviewer #1: (No Response)

Reviewer #3: Yes

6. Review Comments to the Author

Reviewer #1: The authors have addressed my all queries from the first review and revised the manuscript quite extensively. Additional information on QRDR genes and E. coli pathogroups has been provided and discussed well. Overall, the manuscript has vastly improved compared to the previous version.

Reviewer #3: (No Response)

7. PLOS authors have the option to publish the peer review history of their article (what does this mean?). If published, this will include your full peer review and any attached files.

Reviewer #1: No

Reviewer #3: No

---

## [Editor Report · Acceptance letter]

1 Mar 2022

PONE-D-21-27273R1 

Pan-genome and resistome analysis of extended-spectrum ß-lactamase-producing *Escherichia coli*: a multi-setting epidemiological surveillance study from Malaysia 

Dear Dr. Dwiyanto:

I'm pleased to inform you that your manuscript has been deemed suitable for publication in PLOS ONE. Congratulations! Your manuscript is now with our production department. 

Kind regards, 

on behalf of

Dr. Iddya Karunasagar 

Academic Editor

PLOS ONE